# Generation of multitissue cell-cultivated meat via multidirectional differentiation of stable porcine epiblast stem cells

Yixuan Yao[1,5], Gaoxiang Zhu[1,5], Minglei Zhi [1,5], Runbo Li[1,5], Lun Qin[2], Yu Zhang[3], Yachun Chen[3], Xinze Chen[1], Tong Wang[1], Linzi Li[4], Yingjie Wang[1], Shunxin Wang[1], He Zhang[1], Xianchao Feng [4], Aijin Ma[3] ✉, Suying Cao[2] ✉ & Jianyong Han [1] ✉

Cell-cultivated meat has emerged as an alternative approach for sustainable protein production, but replicating the texture and complex composition of conventional meat remains a critical challenge. Here, we developed an efficient approach to produce multitissue engineered meat by leveraging a serum- and animal-component-free differentiation system to direct porcine pregastrulation epiblast stem cells toward muscle, adipose, and endothelium with synergistic functions. These three types of cell progenitors demonstrated autonomous intercellular recognition and coculture compatibility in a scaffold-free 3D suspension system, enabling the spontaneous formation of tissue-mimetic spheroids with enhanced cellular multiplication efficiency. The products of multitissue cultivated meat recapitulated the textural properties of conventional pork and allowed for nutritional modulation. This platform overcomes key limitations in currently employed cultured meat technologies by integrating scalable 3D suspension culture with serum-free, species-specific stem cell multidirectional differentiation, providing an avenue for the development of multitissue cell-cultivated meat.

Cell-cultivated meat (CM) is a food product made by culturing animal cells in vitro to replicate the sensory and nutritional qualities of conventional meat. It has significant potential in terms of sustainability, animal welfare, public health, and related aspects[1,2]. Since a burger patty was cultivated from bovine satellite cells for the first time in 2013[3], CM products have now begun to enter the market. Currently, the research and development of CM still face numerous challenges, including the development of a serum-free differentiation system, multitissue culture, and large-scale cell expansion[4–7]. Moreover, the sensory characteristics and nutritional value of CM have garnered increasing attention from both researchers and consumers[7].

Adult stem cells (ASCs), such as muscle stem cells (MuSCs)[8,9], adipose-derived stem cells (ADSCs)[10,11], fibroadipogenic progenitors (FAPs)[12], and fibroblasts[13], are the more commonly employed cell sources for CM production. However, ASCs invariably lose their proliferation and differentiation capacity during long-term culture, necessitating continuous extraction from animal sources, and immortalized cell lines could pose potential safety risks[4]. Pluripotent stem cells (PSCs) possess the potential for multidirectional differentiation, enabling them to generate all types of cells that constitute meat. Furthermore, PSCs also provide a virtually infinite cellular source owing to their inherent safe and controllable self-renewal capabilities[14],

[1]State Key Laboratory of Animal Biotech Breeding, Frontiers Science Center for Molecular Design Breeding (MOE), China Agricultural University, Beijing, People's Republic of China. [2]Animal Science and Technology College, Beijing University of Agriculture, Beijing, People's Republic of China. [3]School of Food and Health, Beijing Technology and Business University, Beijing, China. [4]College of Food Science and Engineering, Northwest A&F University, Yangling, Shanxi, China. [5]These authors contributed equally: Yixuan Yao, Gaoxiang Zhu, Minglei Zhi, Runbo Li. ✉e-mail: maaj@btbu.edu.cn; 20137602@bua.edu.cn; hanjy@cau.edu.cn

potentially enabling the creation of cell banks and ultimately eliminating the necessity for animal tissue biopsies as material sources[15]. These characteristics render them ideal candidates for use as chassis seed cells in CM production.

Natural meat comprises a diverse array of cells and tissues, including muscle, fat, blood vessels, nerves, fibrous tissue, and potentially resident immune cells[16]. To simulate natural meat effectively, developing a variety of tissues alongside appropriate nutrient profiles is imperative. Three-dimensional (3D) scaffolds and bioprinting have been investigated to replicate the structure of meat by assembling various types of cells[17,18]. However, the isolation of cells from animals to establish the culture, and the implementation of scaffolds, involves costs and complex steps, which are not conducive to large-scale expansion culture. Additionally, microcarrier technologies have also been employed in the development of CM[19,20], but the maximum density that cells can achieve and the cost remain limited.

Here, we successfully induced porcine pregastrulation epiblast stem cells (pgEpiSCs) to undergo directed lineage-specific differentiation into muscle, adipose, and vascular endothelial cells. We also established a scalable 3D suspension culture platform to facilitate intercellular recognition and coculture of three types of cell progenitors through scaffold-free self-assembly. The multitissue CM we generated was similar to real meat in terms of textural properties and allowed for nutritional modulation. Moreover, the multitissue CM demonstrated superiority in terms of springiness and contained a relatively high level of polyunsaturated fatty acids (PUFAs). Our research achieved multilineage differentiation of PSCs, enabling scalable 3D suspension culture for CM production via self-assembly. This advancement provides possibilities for the creation of nutritionally customized and ecologically sustainable meat products.

## Results

### A serum-free approach for adipogenic differentiation from pgEpiSCs

Adipose tissue in mammals is mainly formed by mesenchymal stem cells (MSCs) derived from the mesoderm that undergo lineage specification[21]. For adipocyte differentiation, we employed a serum-free and stage-specific directed differentiation strategy, first guiding pgEpiSCs toward mesenchymal-like cells (adipose progenitor cells), followed by adipogenic induction (Fig. 1a and Supplementary Fig. 1a). To initiate mesoderm differentiation, we activated the WNT/β-catenin pathway with CHIR99021 and inhibited TGF-β signaling using SB431542, as these pathways are critical for mesodermal lineage commitment[22]. The optimal mesoderm induction period was determined to be 3 days on the basis of the maximum reduction in the expression of pluripotency markers (NANOG, OCT4, SOX2) and the upregulation of the expression of the mesoderm-specific marker T (Brachyury), as quantified by qRT-PCR (Supplementary Fig. 1b, c). The loss of pluripotency was confirmed by alkaline phosphatase (AP) staining (Supplementary Fig. 1d) and NANOG downregulation (Fig. 1b), whereas T (Brachyury) immunostaining (Fig. 1b) confirmed successful induction of mesoderm differentiation.

To further initiate adipocyte progenitor cell (APC) differentiation, CHIR99021, LDN193189, FGF2, and EGF were added to promote epithelial-mesenchymal transition (EMT)[23,24]. From Day 3 to Day 12, the expression of APC markers (PDGFRA, ZNF423, and VIM) increased, whereas that of DLK1[25] (an adipogenesis inhibitor) decreased, indicating preadipocyte commitment (Fig. 1c). An additional 6 days were required for the maintenance of APC-stage cells and subsequent cell expansion and proliferation. (Supplementary Fig. 1e). pgEpiSC-derived APCs (pgAPCs) exhibit MSC characteristics, as evidenced by the expression of the PDGFRA⁺CD29⁺CD45⁻ (Fig. 1d) and the genes ITGB1, THY1, ENG, notably, they lack expression of PECAM1, and PTPRC (Fig. 1d, e), which is consistent with the findings of in vivo experiments[26]. To promote adipogenesis, the use of cocktails and PPARγ agonist rosiglitazone was employed. After another 6 days of culture, the master regulators of adipogenesis[27,28] (PPARG, CEBPA) reached peak expression, and the expression of maturation genes[29] (FABP4, LPL) gradually increased (Supplementary Fig. 1f). Upon further maintenance culture, Oil Red O, BODIPY, and Nile red staining confirmed robust lipid droplet (LD) accumulation in pgEpiSC-derived adipocytes (pgADs), with positive expression of the adipocyte maturation marker FABP4[29] (Fig. 1f–h and Supplementary Fig. 1g). Chromosomal stability was maintained throughout differentiation, as evidenced by normal karyotype analysis (Supplementary Fig. 1h).

To verify the adipogenic differentiation process, we compared pgADs with adipocytes derived from porcine FAPs (pFAPs) isolated from muscle tissue (Supplementary Fig. 2a). pFAPs at passage 13 presented diminished adipogenic potential, as indicated by reduced LD formation as well as reduced expression levels of adipogenesis-related genes (Supplementary Fig. 2b–d). RNA-Seq analysis revealed that pgADs presented upregulated expression of key transcription factors (PPARG, CEBPA, ADD1) and lipid metabolism-related genes[30] (PLIN3, AGPAT2) compared with undifferentiated controls (p < 0.01; Fig. 1i). GO enrichment analysis further highlighted the adipogenic differentiation pathway (Fig. 1j). Furthermore, compared to pFAPs, principal component analysis (PCA) revealed correlations among the four cell groups (Supplementary Fig. 2e). Adipose cells derived from both pFAPs and pgEpiSCs were enriched in adipose development-related functions. These adipose cell groups exhibited similar gene expression profiles associated with adipogenic development during differentiation (Supplementary Fig. 2f), suggesting that adipose cells derived from pgEpiSCs share certain characteristics with those from pFAPs. These findings demonstrate that pgEpiSCs could serve as a stable, serum-free source for the generation of functional adipocytes, overcoming the limitations of pFAPs in long-term culture.

### A serum-free approach for vascular cells differentiation from pgEpiSCs

Endothelial cells (ECs) and pericytes are important components of the vascular system that originate from the lateral and posterior mesoderm. To generate vascular cell types from pgEpiSCs, we optimized a serum-free, monolayer stepwise differentiation protocol on the basis of previous studies of vascular cell differentiation[31,32] (Fig. 2a, b). For efficient mesoderm induction, we activated the WNT/β-catenin pathway using CHIR99021 and supplemented the culture with activin A, bone morphogenetic protein 4 (BMP4), and vascular endothelial growth factor (VEGF), which are critical for vascular lineage specification[33,34]. Temporal analysis of pluripotency and germ layer marker genes revealed that the expression of mesodermal genes (T) peaked on Day 1 of differentiation, confirming successful mesoderm induction (Fig. 2c, d). Subsequent optimization identified 5 days of SB431542 and VEGF supplementation as the optimal conditions for vascular lineage specification (Fig. 2e). Further differentiation with VEGF and FGF2 yielded a heterogeneous cell population exhibiting polygonal and fusiform morphologies (Fig. 2b), corresponding to ECs and pericytes, respectively. Immunostaining confirmed the coexistence of CD31⁺eNOS⁺ ECs and αSMA⁺ pericytes (Fig. 2f, g).

Functional validation demonstrated that pgEpiSC-derived vascular cells (pgVCs) formed structurally coherent tube-like networks after 6 h of incubation on a thick layer of Matrigel (Fig. 2h) and exhibited LDL uptake, a hallmark of functional ECs[35] (Fig. 2i). Additionally, these cells maintained >96% viability (calcein and PI assays) and retained a normal karyotype (Fig. 2j–l) throughout the differentiation process. Compared to pgEpiSCs cells, differential analysis revealed that 3044 genes were upregulated following vasculogenesis induction. GO enrichment analysis identified biological processes associated with vasculogenesis and vascular development (Fig. 2m, n). Vascular cell types were successfully generated from pgEpiSCs using this serum-free differentiation system, providing a platform for the

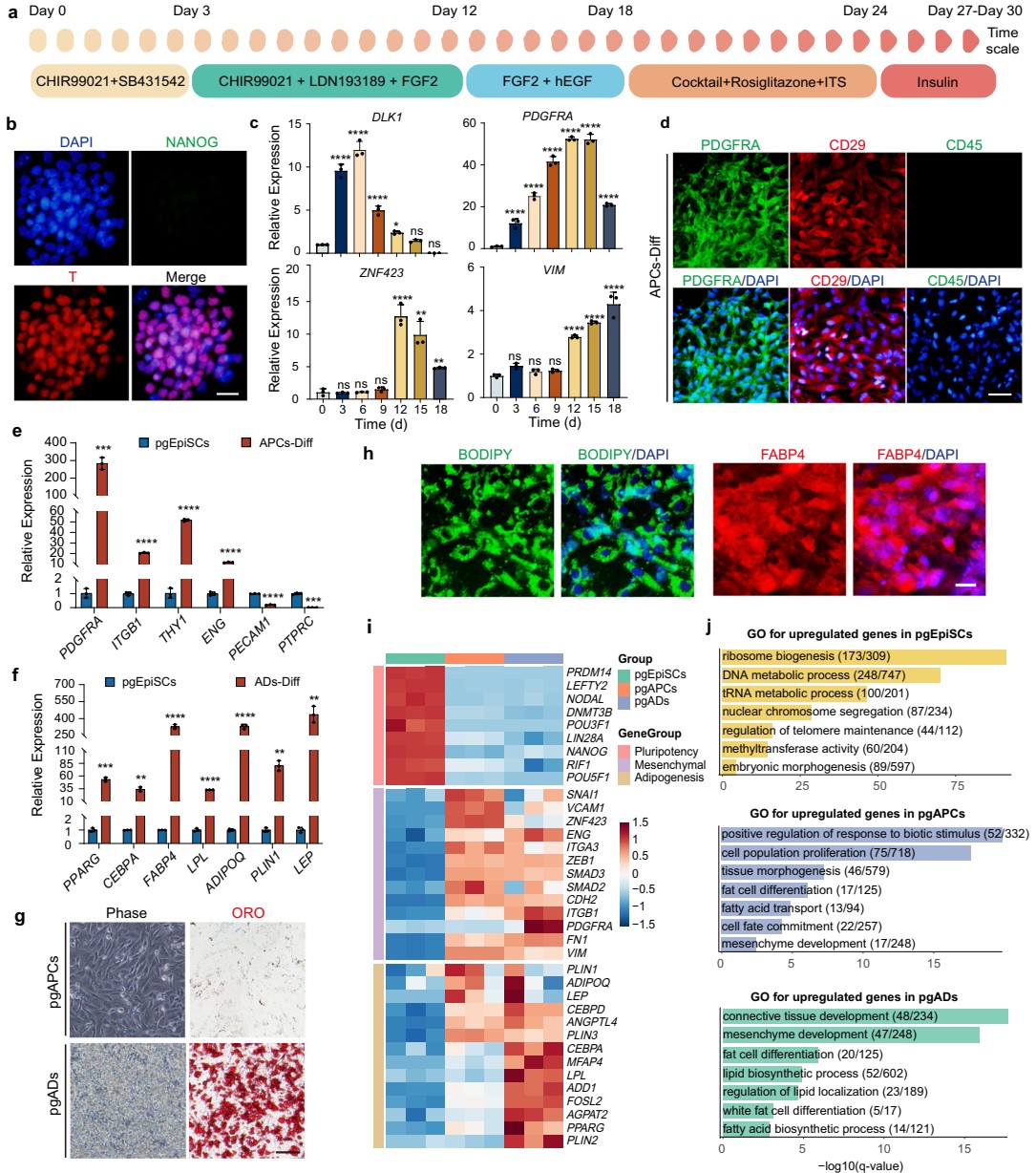

**Fig. 1 | A schematic diagram of adipocyte differentiation from pgEpiSCs in a serum-free culture system. a** Schematic illustration of the differentiation of pgEpiSCs into adipose cells. **b** Immunostaining of NANOG and T in the presence of WNT activation and TGF-β inhibition. Scale bar, 20 μm. **c** Expression of genes related to APC differentiation from Day 0 to Day 18. **d** Immunostaining of PDGFRA, CD29, and CD45 in pgAPCs. Scale bar, 50 μm. **e** pgAPC-expressed genes related to the characteristics of MSCs after 12 days. APCs-Diff, pgEpiSCs differentiated into APCs. **f** Expression of genes related to early (*PPARG*, *CEBPA*) and late (*FABP4*, *LPL*, *ADIPOQ*, *PLIN1*, *LEP*) adipogenic differentiation. ADs-Diff, pgEpiSCs differentiated into adipocytes. **g** Cell morphology and Oil Red O staining before and after the adipogenic differentiation of pgAPCs. Scale bar, 50 μm. **h** Fluorescently stained

image of FABP4 and BODIPY in pgADs. Scale bar, 20 μm. **i** During the process of adipogenesis in pgEpiSCs in vitro, a heatmap representing each cell population revealed similar expression patterns for a specific cluster of genes. **j** Enriched gene ontology (GO) terms in representative clusters that had high *q*-values in different cell populations during adipogenic differentiation of pgEpiSCs. For (**c**, **e**, **f**), error bars indicate ± SD, *n* = 3. *\*p* < 0.05, \*\**p* < 0.01, \*\*\**p* < 0.001, \*\*\*\**p* < 0.0001, and ns indicates *p* ≥ 0.05. One-way ANOVA, followed by Dunnett's multiple comparisons test, was used in (**c**), and a two-tailed Student's *t*-test was used to analyze in (**e**, **f**). Exact *P*-values are listed in Source Data Fig. 1. For (**b**–**h**), similar results were obtained in three independent experiments. Source data are provided as a Source Data file.

subsequent aggregation of multiple lineages and development of a multitissue construct.

## Compatibility of pgEpiSC-derived myogenic and adipogenic cells in 2D and 3D coculture systems

To replicate the nutritional value and structural composition of meat, which is primarily derived from skeletal muscle and adipose tissues, we established a coculture system integrating pgEpiSC-derived myoblasts

and preadipocytes, building on previously optimized myogenic[36] and adipogenic differentiation protocols (Fig. 3a).

In 2D monolayer cocultures, two types of cells were first expanded in growth medium for 2 days until cell fusion and a density of 80% were achieved, followed by differentiation induction. Optimization of the culture protocol revealed that a 1:1 ratio of pgEpiSC-derived myoblasts to preadipocytes supported codifferentiation into mature cell types (Supplementary Fig. 3a–c). Brightfield microscopy shows a mixed

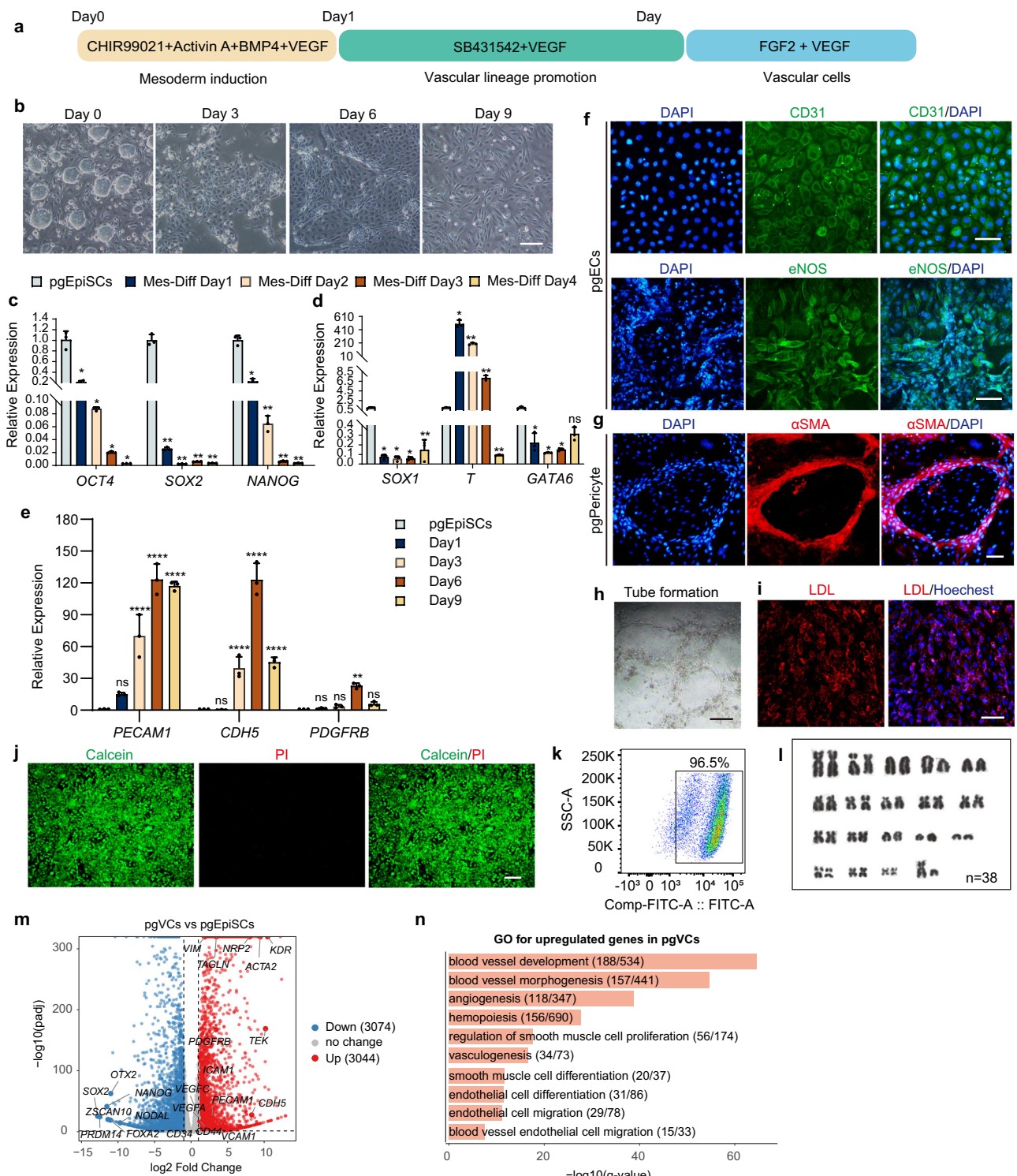

distribution of adipocytes and skeletal muscle fibers (SMFs) (Fig. 3b). Oil Red O and immunofluorescence staining confirmed the maturation of adipocytes and muscle cells (Fig. 3c–e). qRT-PCR analysis further confirmed that two cell types were capable of codifferentiating into mature forms: SMFs (*MYOG, MYH2, MYH3*) (Fig. 3f) and adipocytes (*ADIPOQ, PLIN1, LEP*) (Fig. 3g). Notably, Medium 1 demonstrated superior performance in supporting this differentiation process. These results demonstrated the compatibility of pgEpiSC-derived myoblasts and preadipocytes in 2D cocultures.

To simulate in vivo tissue development, we established edible 3D scaffolds, as pgEpiSC-derived myoblasts had previously been shown to

grow and differentiate on such matrices[36]. pgEpiSC-derived preadipocytes were first seeded on scaffolds to confirm their survival and differentiation potential (Supplementary Fig. 4a). After 10 days of culture, the cells exhibited high viability (>90%) and differentiated into adipocytes with abundant LDs (Supplementary Fig. 4b–f). Adipogenic maturation was further supported by elevated mRNA levels of *FABP4, ADIPOQ, LEP,* and *PLIN1*, whereas pluripotency genes (*OCT4, NANOG*) were downregulated (Supplementary Fig. 4g).

To establish the 3D scaffold, pgEpiSC-derived myoblasts and preadipocytes were mixed at a 1:1 ratio and seeded onto scaffolds (Fig. 3h). The cells adhered efficiently to the scaffold surface and

**Fig. 2 | Establishment of a protocol for the differentiation of vascular endothelial cells from pgEpiSCs in a serum-free culture system. a** Schematic illustration of the committed induction process of pgEpiSCs into vascular cell type. **b** Cellular morphological observation of the differentiation induction process. Scale bar, 50 μm. **c**, **d** Expression of genes related to pluripotency (*OCT4, SOX2, NANOG*), ectoderm (*SOX1*), mesoderm (*T (Brachyury)*), and endoderm (*GATA6*). **e** Expression of genes related to ECs (*PECAM1, CDH5*) and pericytes (*PDGFRB*). **f**, **g** Immunostaining of pgECs (CD31 and eNOS) and pgPericytes (αSMA). Scale bar, 50 μm. pgECs: pgEpiSC-derived ECs, pgPericytes: pgEpiSC-derived pericytes. **h** Tube formation from pgVCs. Scale bar, 50 μm. **i** LDL uptake assessment of pgVCs. Scale bar, 50 μm. **j** Representative images of calcein-AM (live, green) and PI (dead, red) staining of pgVCs. Scale bar, 200 μm. **k** Flow cytometric assay for the survival of pgVCs. **l** Karyotype analysis of pgEpiSCs after vascular cell differentiation, which maintains normal chromosome numbers (*n* = 38). **m** Volcano plot of differentially expressed genes between pgEpiSCs and pgVCs. Statistical significance was assessed using a two-sided Wald test, with *P*-values adjusted for multiple testing using the Benjamini–Hochberg method. Genes with an adjusted *P*-value < 0.05 and |log2 fold change| ≥ 1 were considered significantly differentially expressed. **n** GO terms in representative clusters that had high *q*-values in different cell populations during differentiation of pgEpiSCs into vascular cells. For (**c**–**e**), the error bars indicate the means ± SDs, *n* = 3. *$p < 0.05$, **$p < 0.01$, ****$p < 0.0001$, and ns indicates $p > 0.05$. Similar results were obtained in three independent experiments and represent a significant difference using Two-way ANOVA, followed by Dunnett's multiple comparisons test. Exact *P*-values are listed in Source Data Fig. 2. For (**b**–**l**), similar results were obtained in three independent experiments. Source data are provided as a Source Data file. Mes-Diff, pgEpiSCs differentiated into mesoderm.

maintained >92% viability (Fig. 3i, j), and after 10 days of culture, populated scaffold pores formed a tissue-like structure (Fig. 3k and Supplementary Fig. 4h). Immunofluorescence and qRT-PCR confirmed the coexistence of SMFs and adipocytes, with lipid accumulation reaching high levels (Fig. 3l–n). These coculture systems in 2D and 3D environments enabled the generation of a multitissue CM with both skeletal muscle and adipose components. However, both 2D monolayers and scaffold-based systems exhibit limitations due to their smaller surface area to volume ratio, prompting the development of new culture systems for large-scale cell expansion.

### 3D suspension culture enables long-term expansion of pgEpiSC-derived progenitors

To overcome the limitations of 2D and scaffold-based systems, we explored 3D suspension culture for long-term in vitro expansion of early progenitor populations (pgEpiSC-derived progenitor cells (pgMPCs) and pgAPCs). Using a nonadherent-suspension-shaking system[37] (Fig. 4a), we cultivated pgMPCs (MYOD+, PAX7+) and pgAPCs (CD29+, PDGFRA+) separately (Supplementary Fig. 5a–d). From the first day of culture, the cells self-aggregated into spheroids, with volumes increasing progressively over time (Fig. 4b–e). The spheroids reached diameters of 800–1200 μm while maintaining high viability (Supplementary Fig. 5e, f). Morphological analysis revealed distinct features: muscle spheroids (Fig. 4f) presented smooth surfaces with tightly packed cells, whereas adipose spheroids (Fig. 4g) presented rough, irregular surfaces with a looser cell arrangement.

Functionally, muscle spheroids developed a fibrous tissue-like interior and expressed myofiber differentiation markers such as myosin (Fig. 4h, i), whereas adipose spheroids were predominantly occupied by LDs and expressed FABP4 (Fig. 4j, k). Both spheroid types expressed Collagen IV (Supplementary Fig. 5g, h), suggesting that these spheroids were involved in extracellular matrix (ECM) organization[38]. Transcriptomic profiling and qPCR analysis revealed that 3D suspension culture of muscle progenitors significantly upregulated genes associated with skeletal muscle development and ECM formation compared to 2D conditions, as evidenced by elevated expression of key myogenic markers such as *MYOG* and *MYH3* (Fig. 4l, m and Supplementary Fig. 5i). While the expression of adipogenic markers (*PPARG, LPL*) in 3D adipocytes showed moderate differences compared to 2D conditions, the 3D system exhibited superior performance in tissue morphogenesis and cell recognition (Fig. 4n, o and Supplementary Fig. 5j).

We further tested the feasibility of 3D suspension culture for vascular progenitor cells (VPCs) (Supplementary Fig. 5k–p). Vascular spheroids formed rough surfaces and irregular cell arrangements and expressed markers of ECs (*PECAM1, CDH5*) and pericytes (*PDGFRB*). However, their proliferative capacity was limited after 10 days of cultivation, with minimal expansion thereafter (Supplementary Fig. 5k, l).

These findings demonstrate that 3D suspension culture systems outperform 2D systems in supporting the long-term expansion, cell differentiation, or tissue morphogenesis of muscle and adipose progenitors. This approach may provide a scalable platform for generating multitissue CM with defined ratios of skeletal muscle, adipose, and vascular components.

### 3D multilineage cocultures generate multitissue CM with enhanced nutritional and structural properties

Next, we designed a multilineage coculture protocol to enable the self-assembly of muscle, adipose, and vascular progenitors into tissue-like spheroids and further recapitulate complex tissue architectures and improve the fidelity of meat-like constructs.

Dual-lineage coculture of pgMPCs and pgAPCs at a 1:1 ratio ($5 \times 10^5$ cells/mL) in 3D suspension culture yielded spheroids by Day 1 (Supplementary Fig. 6a, b). After 5 days of proliferation in differentiation medium, the spheroids were transferred to terminal differentiation conditions for 9 additional days and maintained until Day 30. During this period, spheroids increased in size to ~1000 μm (Supplementary Fig. 6c) while maintaining high viability (>90%) (Supplementary Fig. 6d, e), with cells exhibiting random aggregation (Supplementary Fig. 6f, g) and progressively developing a denser, more organized architecture and distinct tissue domains compared to those observed on Day 15 (Supplementary Fig. 6h, i). Functional validation revealed high LD accumulation and coexpression of SMF-specific proteins (Supplementary Figs. 6j, 7a, b). qPCR analysis confirmed the upregulation of genes associated with myogenic maturation (*MYOG, MYH2, MYH3*), adipocyte development (*ADIPOQ, LEP, PLIN1*), and ECM formation (*COL3A1, COL5A2*) (Supplementary Fig. 7c). In addition, the efficiency of muscle, adipose, and ECM formation in the coculture spheroids was intermediate compared with that observed in spheroids derived from individual cell types (Supplementary Fig. 7d). Moreover, the final tissue composition (muscle-to-fat ratio) could be controllable through seeding ratios (Supplementary Fig. 8).

Tri-lineage coculture of pgMPCs, pgAPCs, and pgVPCs at a 1:1:1 ratio further increased structural complexity (Fig. 5a and Supplementary Fig. 9a). Over time, the diameter of the spheroids progressively increased, reaching approximately 1,000 μm by Day 30 with high cell viability (Fig. 5b, c, Supplementary Fig. 9b, c), while cell density reached $1.35 \times 10^8$/mL (Fig. 5d), resulting in a denser and more structurally complex tissue architecture (Fig. 5e and Supplementary Fig. 9d) with random aggregation (Supplementary Fig. 9e, f). Notably, spheroids began forming internal cavities on Day 10, after which they expanded over time (Fig. 5b). Multiple tissue types were identified within the spheroids (Fig. 5e, f), and immunostaining confirmed the coexistence of LDs (BODIPY), SMFs (myosin), and ECs (CD31) within the spheroids (Fig. 5g, Supplementary Fig. 9g, h), accompanied by robust ECM production (Supplementary Fig. 9i). qPCR analyses revealed simultaneous upregulation of the following lineage-specific markers: myogenic maturation-related genes (*MYOG, MYH2, MYH3*), adipocyte markers (*ADIPOQ, LEP*), vascular markers (*PECAM1, CDH5,*

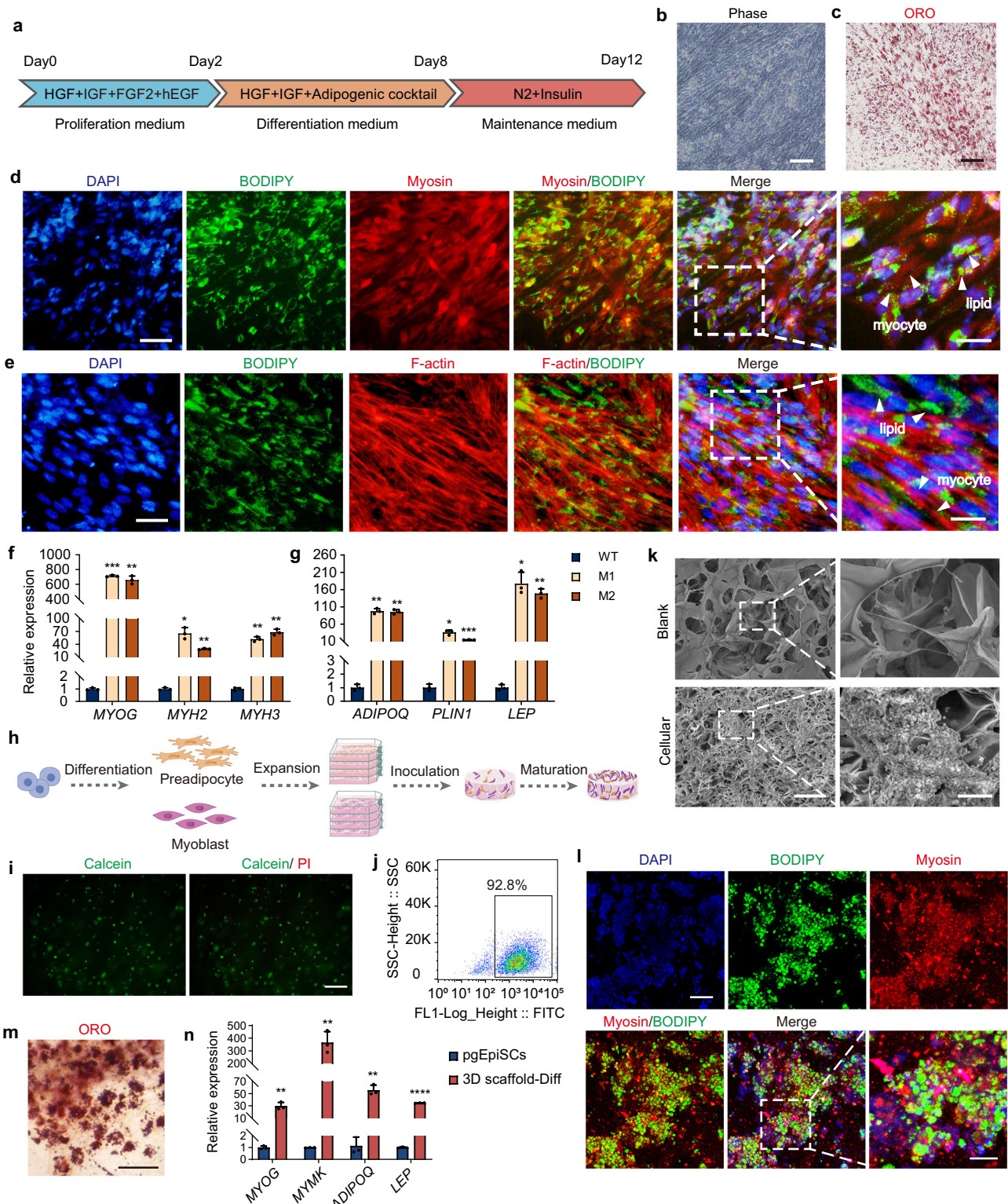

*PDGFRB*), and ECM-related genes (*COL3A1, COL5A2*) (Supplementary Fig. 9j). Transcriptomic profiling revealed that multitissue spheroids simultaneously exhibited characteristics of muscle, adipose, and vascular tissues. Notably, the expression levels of many marker genes related to muscle (*DES, ACTN2, MYL3*), adipose (*CEBPD, LPL*), and ECM (*FN1, COL3A1*) in multitissue spheroids were even greater than those in spheroids derived from a single-cell type (Fig. 5h, Supplementary Fig. 9k–n). These findings indicate that coculturing multiple cell types promotes their respective differentiation processes. Furthermore, the final tissue composition, particularly the ratio of muscle to fat to endothelium, could be modulated by adjusting the initial cell seeding ratios (Supplementary Fig. 10a–c).

The analysis of nutritional composition in multitissue spheroids demonstrated that the relative proportions of amino acids closely resemble those in native meat, albeit with a lower total amino acid (TAA) content. In contrast, multitissue spheroids exhibited higher total fatty acid (TFA) levels, accompanied by an elevated proportion of polyunsaturated fatty acids (PUFAs). Notably, the ∑PUFA/∑SFA ratio

**Fig. 3 | pgEpiSC-derived myogenic and adipogenic cells were cocultured in 2D and 3D systems. a** Schematic illustration of the committed induction process of cocultured pgEpiSC-derived myoblasts and preadipocytes. The cells were allowed to proliferate for 2 days, cultured in differentiation medium for 6 days, and subsequently maintained in maintenance medium for an additional 4 days. **b, c** Brightfield and Oil Red O staining of 1: 1 cocultured cells. Scale bar, 50 μm. **d, e** Immunostaining of myotubes and LDs from 1: 1 cocultured cells. The images were taken using confocal microscopy after immunofluorescence staining of mature SMFs with myosin or F-actin, LDs with BODIPY, and nuclei with DAPI. Scale bar, 50 μm (left) and 20 μm (right). **f, g** Expression of genes related to the maturation of myogenic differentiation (*MYOG, MYH2, MYH3*) and adipogenic differentiation (*ADIPOQ, PLIN1, LEP*) in 2D cocultured cells. M1, medium 1; M2, medium 2. Error bars indicate means ± SD, $n = 3$. *$p < 0.05$, **$p < 0.01$, ***$p < 0.001$. Similar results were obtained in three independent experiments and represent a significant difference using Two-way ANOVA, followed by Dunnett's multiple comparisons test. Exact *P*-values are listed in Source Data Fig. 3. **h** Schematic diagram of the inoculation of pgEpiSCs differentiated into pgEpiSC-derived myoblasts and preadipocytes, subsequently inoculated on plant-based edible 3D scaffolds. **i** The cells were stained with calcein-AM (green) and PI (red) after inoculation into the scaffold culture for 24 h. Scale bar, 100 μm. **j** Flow cytometric assay for the survival of cocultured cells on 3D scaffolds for 24 h. **k** Empty scaffolds and scaffolds with cocultured cells inoculated for 12 days were observed using SEM. Scale bar, 200 μm (left) and 20 μm (right). **l** Immunostaining of myotubes and LDs from cocultured cells with scaffolds. Scale bar, 50 μm. **m** Oil Red O staining of LDs from cocultured cells with scaffolds. Scale bar, 50 μm. **n** Expression of genes related to the maturation of myogenic differentiation (*MYOG, MYMK*) and adipogenic differentiation (*ADIPOQ, LEP*) in cocultured cells with scaffolds. Error bars indicate means ± SD, $n = 3$. **$p < 0.01$, ****$p < 0.0001$. Similar results were obtained in three independent experiments and represent a significant result using a two-tailed Student's *t*-test. Exact *P*-values are listed in Source Data Fig. 3. For (**b**–**h**) and (**i**–**n**), similar results were obtained in three independent experiments. Source data are provided as a Source Data file.

---

was approximately twofold higher than that of native meat (Fig. 5i, j and Supplementary Fig. 10d–g). The results of Gene Set Enrichment Analysis (GSEA) showed that the signaling pathway related to unsaturated fatty acid (UFA) biosynthesis was significantly activated in multitissue spheroids compared with pig muscle tissue (Fig. 5k). An enrichment analysis was performed on these genes (Fig. 5l), which are implicated in the biosynthesis of long-chain fatty acids and the regulation of lipid metabolism[39–41]. After being processed into sausages, pgCM sausages exhibited a comparable appearance to both conventional pork sausages and pgEpiSCs sausages (Fig. 5m). Textural profile analysis (TPA) revealed that pgCM exhibited similar hardness, cohesiveness, gumminess, chewiness, and resilience to conventional pork sausages, with all these attributes outperforming those of pgEpiSCs-derived sausages. Notably, both pgCM sausages and pgEpiSCs-derived sausages showed better springiness, which may be attributed to the intrinsic cellular properties (Fig. 5n). And compared to pork sausages, pgCM sausages show a higher relative abundance of aldehydes and alcohols, whereas the levels of other compound classes are lower. The most pronounced differences are found in aldehyde and hydrocarbon content, suggesting distinct volatile profiles that may influence sensory characteristics (Supplementary Fig. 10h). These results demonstrate that 3D suspension coculture of multilineage progenitors from pgEpiSCs enables the large-scale production of multitissue CM. Compared to native meat, CM exhibits a similar textural profile, while its nutritional composition, although distinct, could be controlled by modulating the initial cell seeding ratios.

## Discussion

CM, as a sustainable and eco-friendly future meat product, is attracting increasing attention. However, achieving the tissue composition and nutrient richness of conventional meat products remains challenging[42–44]. In this study, we achieved the directed differentiation of pgEpiSCs into multiple tissue types in a serum-free system and facilitated the self-assembly of multitissue CM using a 3D suspension system, providing possibilities for the creation of nutritionally modulated CM products.

A key challenge in CM development is the limited expandability of initiating cell lines, which are predominantly confined to unipotent cell types—including muscle satellite cells, mesenchymal stem cells (MSCs), adipose-derived stem cells (ADSCs)[45], and fibroblasts. A critical limitation of these unipotent cells is their inability to undergo long-term expansion and maintain differentiation capacity during continuous passaging, necessitating repeated isolation from animal sources, which raises significant concerns regarding animal welfare and sustainability principles. The pluripotency of PSCs endows them with the potential for multitissue differentiation, enabling them to differentiate into all types of cells. Moreover, their self-renewal ability allows for unlimited expansion, rendering them ideal candidate cells

for the development of CM. In our study, we harnessed stem cell resources to elucidate the diverse differentiation mechanisms of pgEpiSCs. We successfully established a stage-specific directed differentiation strategy of a single pgEpiSCs line into skeletal muscle, adipose, and vascular endothelial cells with a serum- and animal-component-free differentiation system. Although the acquisition and long-term cultivation of embryo-derived pluripotent stem cells under feeder-free conditions have not yet been fully achieved, these limitations could be overcome through progressive cell adaptation and optimization of the culture system.

Beyond muscle cells, current research on CM has also investigated the integration of other tissue components. Fat provides essential nutrients and contributes to the flavor, texture, and tenderness of meat, all of which have a significant impact on consumer preferences[46–50]. Furthermore, blood vessels are vital for sustaining tissue growth and structure by supplying oxygen and nutrients in vitro[51,52]. In this study, we investigated the coculture of three cell types derived from pgEpiSCs and demonstrated that different progenitor cell populations could recognize one another and self-assemble into multitissue spheroids, with the final tissue composition being controllable through the initial seeding ratios. The amino acid (AA) and fatty acid (FA) types and compositions of CM we produce are comparable to those of traditional pork. However, the total amino acid content is relatively low, whereas the total fatty acid content is comparatively high. Notably, the ∑PUFA/∑SFA ratio in CM is significantly higher than in traditional pork, indicating a more favorable fatty acid profile, which may be more beneficial for cardiovascular health. The texture and tenderness of CM may be related to factors such as muscle fibers, connective tissues, extracellular matrix proteins, and scaffold materials[7,9,47]. The 3D-cultured multitissue CM in our study not only comprised myofibers, adipose tissue, and microvessels but also generated a significant amount of ECM and collagen. After cooking, the CM exhibited suitable tenderness and elasticity. Although the specific interaction mechanisms between different cells are still unclear, and a certain disparity persists between the maturity of cultured cells and that of native tissues, we have provided a potential technical approach for the self-assembly coculture of multiple cell types. Moreover, in future studies, optimizing culture medium and adjusting the seeding ratio of the initial cell population may enable modulation of the tissue architecture and nutritional profile of CM, bringing it closer to the characteristics of native meat.

2D monolayer culture is limited by its finite surface area and the necessity for continuous passaging, which precludes it from serving as an effective method for large-scale cell expansion. In the present study, the fabrication of CM by integrating 3D scaffolds with cells was the most prevalent approach[9,53,54]. However, the cost of scaffold fabrication, the introduction of exogenous materials, and the persistent limitation in cell expandability remain as bottlenecks. In this study, we

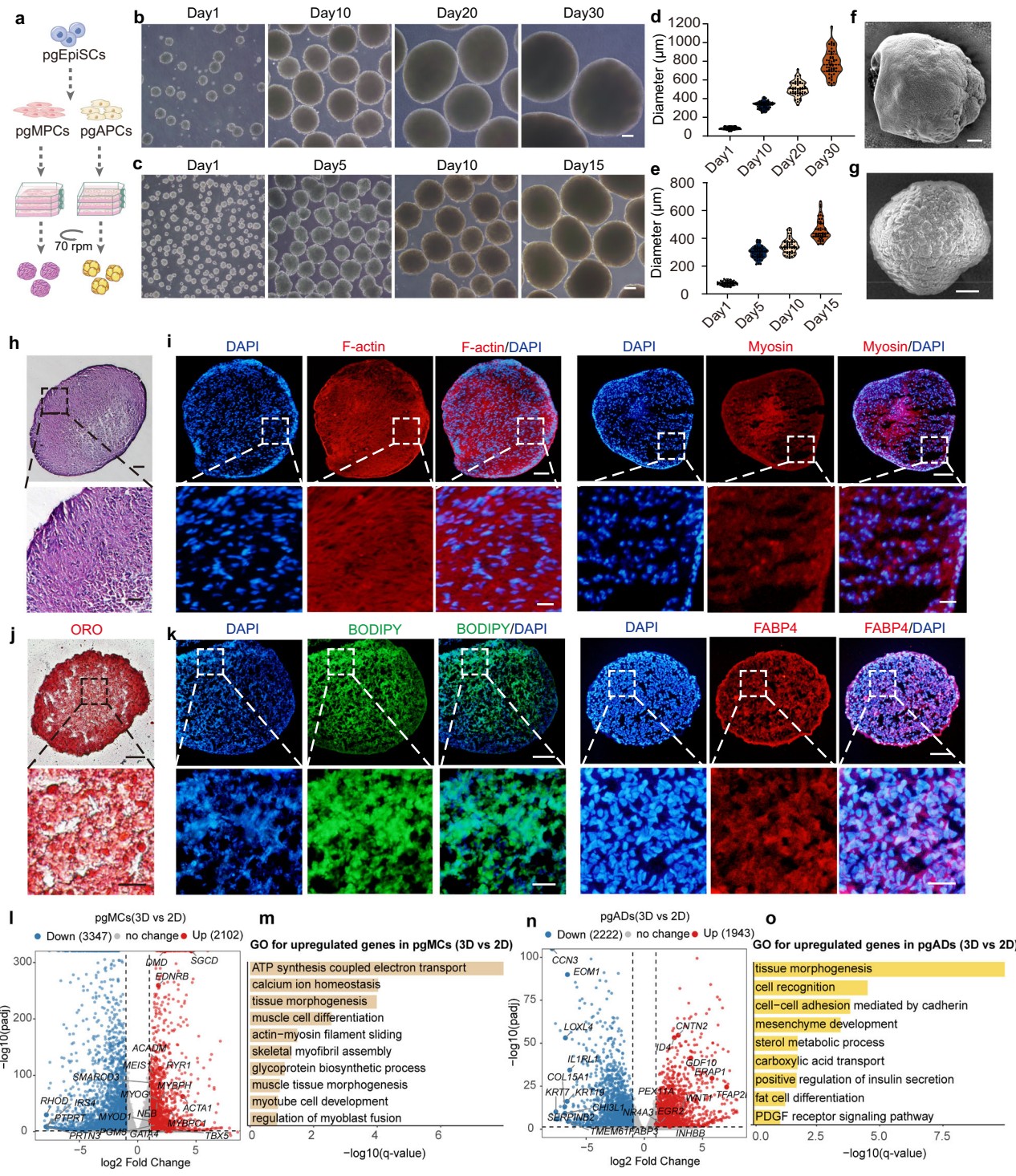

employed a simple yet efficient 3D suspension culture system for the cultivation of multiple cell types, which spontaneously self-assemble into spheroids by relying on their endogenous ECM, forming multitissue architectures that effectively enhanced cellular proliferation efficiency and tissue complexity. The suspension scaffold-free strategy based on the spheroid method not only eliminates the dependence of cells on scaffolds, but also, when combined with a standardized stem cell directed differentiation system, could generate homogeneous multitissue spheroids with compositional complexity and texture characteristics in a repeatable manner. It also facilitates large-scale cell culture and expansion, and helps seamlessly adapt to bioreactor systems. However, the requirement for an initial 2D differentiation step

from pgEpiSCs to progenitor cells prior to 3D coculture results in complex procedures, necessitating further simplification and optimization. Overall, it is expected to overcome the limitations of scalability and repeatability, thereby providing a feasible and industrializable CM production technology approach.

In conclusion, we generated a multitissue CM product via the multilineage differentiation of pgEpiSCs and the application of 3D suspension culture technology, thereby endowing the CM product with textural properties resembling those of native pork and allowing for nutritional modulation. We hope that our study could provide a technological approach for CM development and increase our understanding of the application of stem cells in the food industry.

**Fig. 4 | pgEpiSC-derived progenitors were capable of long-term expansion in 3D suspension culture. a** Schematic representation of the formation of spheroids differentiated into 3D suspensions through a nonadherent-suspension-shaking system. **b** Light microscopy images of spheroids from pgMPCs that formed from Day 1 to Day 30. **c** Brightfield images of spheroids from pgAPCs subjected to 3D adipogenic differentiation for 15 days over the time course. **d, e** Diameter statistics of 3D suspension-differentiated spheroids from pgMPCs and pgAPCs. **f** Muscle spheroids, visualized by scanning electron microscopy (SEM), exhibited smooth surface features. **g** SEM revealed that the surface of the adipose spheroids was rough and that the arrangement of the cells was relatively loose. **h** Representative images of Hematoxylin and eosin (HE)-stained muscle spheroids. **i** Representative fluorescence images of cytoskeletal protein (F-actin), mature muscle fibers (myosin) in muscle spheroids on Day 30. **j** Representative images of Oil Red O staining of adipose spheroids. **k** Representative fluorescence images of LDs (BODIPY) and adipocytes (FABP4) in adipose spheroids on Day 15. **l** Volcano plot of differentially expressed genes between 2D and 3D myogenic differentiation of pgEpiSCs. **m** GO terms in representative clusters that had high $q$-values during differentiation between 2D and 3D pgMCs. **n** Volcano plot of differentially expressed genes between 2D and 3D adipogenic differentiation of pgEpiSCs. **o** GO terms in representative clusters that had high $q$-values during differentiation between 2D and 3D pgADs. For (**b, c**), scale bar, 50 μm. For (**f–h, j**), scale bar, 100 μm. For (**i, k**), scale bars, 50 μm (above) and 20 μm (below). For (**d, e**), $n = 50$, the violin plot is cut at the lowest and highest values, dotted line: quartile, dashed line: median, and dots represent single data points. For (**l, n**), Statistical significance was assessed using a two-sided Wald test, with $P$-values adjusted for multiple testing using the Benjamini–Hochberg method. Genes with an adjusted $P$-value < 0.05 and |log2 fold change| ≥ 1 were considered significantly differentially expressed. For (**b–k**), similar results were obtained in three independent studies. Source data are provided as a Source Data file.

## Methods

### Ethical statement
All the mouse and pig experiments performed were approved by the Institutional Animal Care and Use Committee of China Agricultural University (AW 52114202-3-01).

### Animals
Female ICR mice used for isolation of mouse embryonic fibroblasts (MEFs) were purchased from Beijing SiPeiFu Biotechnology CO., Ltd (Beijing, China). All mice were individually housed under a 12-h light/dark cycle in a sterile environment and provided with food and water ad libitum. The ambient temperature for housing mice in this study was maintained at 21–26 °C, with relative humidity controlled at 50–60%. The feeder cells for pgEpiSCs were prepared from MEFs treated with mitomycin C (Selleckchem, S8146).

Nongda Xiang pig (1-week-old) was used for the isolation of porcine FAPs from the China Agricultural University Experimental Miniature Pig Farm.

### Pig pregastrulation epiblast stem cells (pgEpiSCs) culture medium
pgEpisCs cell lines were derived from the embryo of NongDa Xiang pig in the previous work[14]. The culture medium configuration is described in detail in Supplementary Tables 1, 2. Pig pgEpiSCs were cultured in 3i/LAF culture medium under 20% $O_2$ and 5% $CO_2$ at 37 °C. They were maintained on MEF feeders and passed using Accutase cell dissociation reagent (Gibco, A11105-01) at a ratio from 1:3 to 1:5 every 2–3 days as single cells. The specific number of passages and the dilution ratios should be adjusted according to actual conditions.

### Isolation and culture of porcine FAPs
1.5 g of the longest back muscle was taken and washed three times with DPBS (Gibco, C14190500CP) containing 10% penicillin-streptomycin (PS, Thermo Fisher Scientific, 15140-122). After being minced, collagenase II (Coolaber, CC3791G) stock solution (10×) was added, and the mixture was digested at 37 °C on a shaking incubator set to 90 rpm for 1 h. The digestion process was terminated using complete culture medium (DMEM/F12 (Thermo Fisher Scientific, 10565-018) supplemented with 1% penicillin-streptomycin (PS, Thermo Fisher Scientific, 15140-122), 1% nonessential amino acids (NEAA, Thermo Fisher Scientific, 1140-050), 10% fetal bovine serum (FBS, Gibco, 16000-044), and 5 ng/mL FGF2 (PeproTech, 100-18B)), followed by sequential filtration through 100 μm and then 40 μm cell strainers. The cell suspensions were centrifuged at 1500 × $g$ for 5 min, then resuspended in Red Blood Cell Lysis Buffer (Solarbio, R1010) and incubated on ice for 5 min. Following centrifugation at 1500 × $g$ for 5 min, the cells were resuspended and seeded into gelatin-coated (Stem Cell Technologies, 07903) T75 culture dishes for incubation in a cell culture incubator at 37 °C with 5% $CO_2$. Cells were separated using differential adherence methods; after 2 h, the supernatant was removed and replaced with fresh culture medium. Adherent cells were designated as FAPs. Cell morphology and growth conditions were observed under an inverted microscope, with media changes occurring every 2 days to continue cultivation. While cell fusion reached 90%, the cultured medium was changed to FAPs differentiation medium (DMEM/F12 supplemented with 1% PS, 1% NEAA, 10% FBS, 500 μM 3-isobutyl-1-methylxanthine (IBMX, Sigma, I5879), 1 μM Dexamethasone (Dex, Sigma, D4902), 10 μg/mL insulin (Sigma, I6634)) for 6 days, and FAPs maintenance medium (DMEM/F12 supplemented with 1% PS, 1% MEM NEAA, 10% FBS, 10 μg/mL insulin) for 3 days. The culture medium configuration is shown in Supplementary Tables 1, 2.

### Adipogenic differentiation of pgEpiSCs by serum-free induction
For adipogenic differentiation, pgEpiSCs were dissociated into single cells by Accutase cell dissociation reagent without feeder, then inoculated in a 6-well plate at a density of $2 \times 10^5$ cells/well in feeder-free medium supplemented with 10 μM Y-27632 (Selleckchem, S1049). ADM basic medium (BM) consisted of DMEM/F12 supplemented with 1% PS, 1% NEAA, 0.1 mM β-mercaptoethanol, 15% knockout serum replacement (KOSR), and 200 μM ascorbic acid. In the first stage, cells were switched to ADM I medium consisting of 3 μM CHIR99021 and 10 μM SB431542 for 3 days, then harvested using TrypLE (Gibco, 12605010) and inoculated in ADM II medium consisting of 3 μM CHIR99021, 0.5 μM LDN193189, and 20 ng/mL FGF2 for 9 days during the second stage. Then, the medium was replaced with ADM III for 6 days with 10 ng/mL FGF2 and 10 ng/mL hEGF. When cells reaching 90% confluency, ADM IV, which comprised of 1% ITS, 1 μm Rosiglitazone, and 500 μM IBMX, 1 μM Dex, 10 μg/mL insulin was supplied and maintained for 6 days, aiming for adipogenesis. Finally, the cells were incubated for 4–7 days in ADM V, consisting of 10 μg/mL insulin. The timing of each differentiation stage has been optimized based on tracing the gene expression profiles across stages. The culture medium configuration is shown in Supplementary Tables 1, 2.

### BODIPY staining
The cell samples were washed with DPBS, then fixed in 4% PFA at room temperature for 1 h. Subsequently, the cells were incubated with 1 μg/mL Bodipy493/503 (Duofluor, P10051-100) for 1 h and then washed three times with DPBS. Finally, the nuclei were stained with DAPI for 3 min before being immediately observed and photographed under a fluorescence microscope.

### Oil Red O staining
After adipogenic induction, the cells were washed twice with DPBS and fixed in 4% PFA for 1 h at room temperature. Subsequently, the cells were rinsed three times with double-distilled water (ddH2O). Next, 2 mL of 60% isopropanol was added and incubated at room temperature for 2 min. The isopropanol was then removed, and the cells were

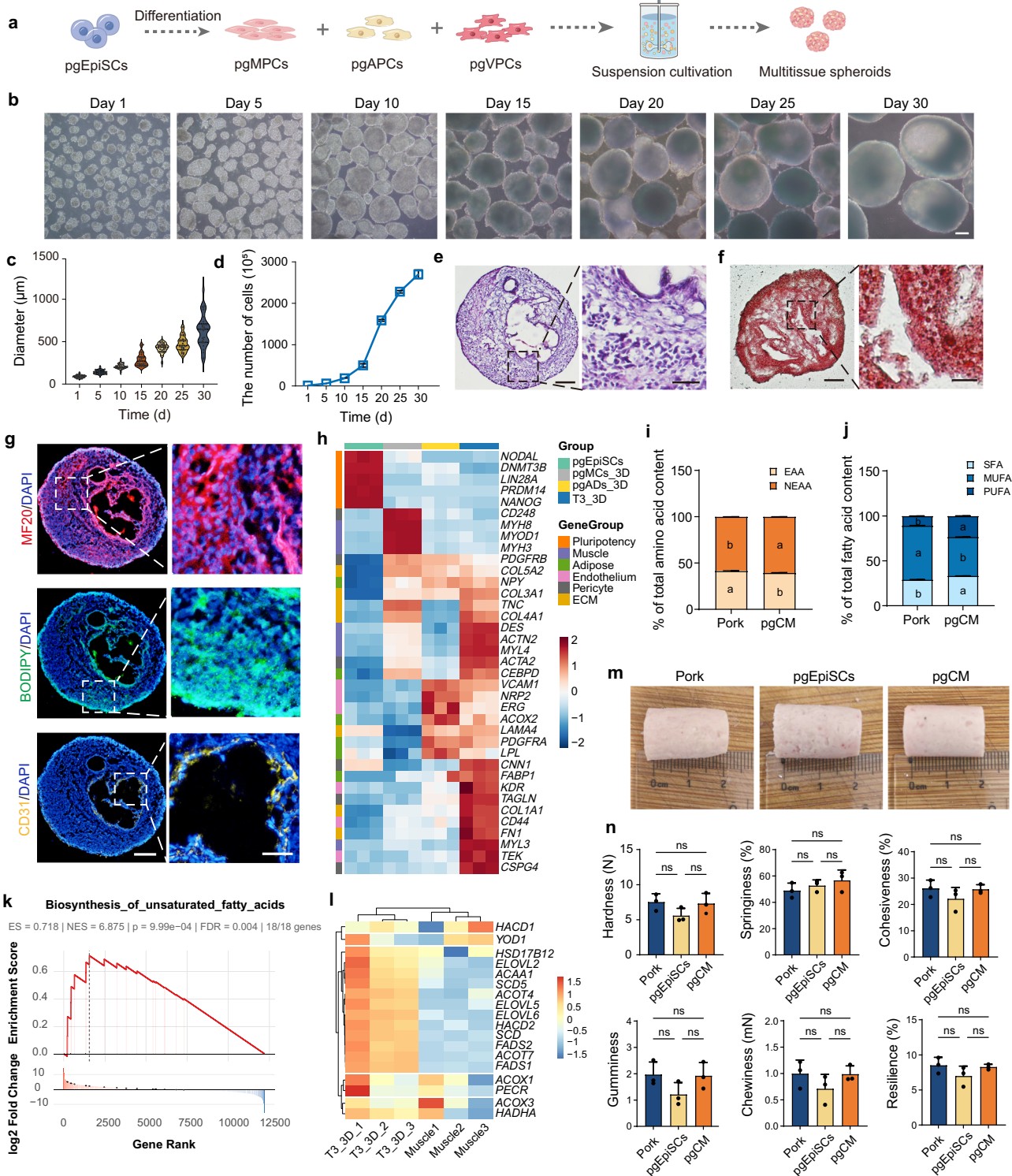

treated with 1 mL of Oil Red O working solution (Oil Red O solution (Sigma-Aldrich, O1391): ddH$_2$O = 3:2) for incubation at room temperature for 10 min, followed by washing with ddH$_2$O until no color remained. Finally, LD formation was observed and documented under an inverted microscope.

## Nile Red staining
The fixed cells were washed three times with DPBS and then stained in the dark with Nile Red (Sigma-Aldrich, 19123). The cells were washed three times with DPBS, and the nuclei were stained with DAPI for 3 min. Fluorescence microscopy followed immediately.

## Vascular cells differentiation of pgEpiSCs by serum-free induction
For monolayer vascular cell differentiation of pgEpiSCs, pgEpiSCs were dissociated into small clumps by Accutase without a feeder, then inoculated in a 6-well plate at a density of $2 \times 10^5$ cells/well in feeder-free medium supplemented with 10 µM Y-27632. BM was consisting of DMEM/F12 supplemented with 1% PS, 1% NEAA, 0.1 mM β-mercaptoethanol, 15% KOSR, and 200 µM ascorbic acid. Small pieces of pgEpiSCs were resuspended in VDM I and maintained for 1 day. For vascular specification from pgEpiSCs, VDM I was replaced with VDM II, and the cells were inoculated for 5 days. Subsequently, the culture

**Fig. 5 | The CM derived from multitissue spheroids was similar to that derived from pork in terms of nutrition and texture. a** Schematic representation of the 3D aggregation model for pgMPCs, pgAPCs, and pgVPCs. **b** Light microscopy images of aggregates formed from various cell sources from day 1 to day 30. Scale bar, 100 μm. **c** Variation in the diameter of 3D-suspended differentiated spheroids. **d** The changes in the cell density of 3D-suspended differentiated spheroids. Representative images of multitissue spheroids observed through **e** HE staining and **f** Oil Red O staining. Scale bar, 100 μm. **g** Immunofluorescence staining of the Day 25 spheroid; MF20, BODIPY, and CD31 are muscle fiber, adipocyte, and EC markers, respectively. Scale bar, 100 μm (left) and 50 μm (right). **h** Heatmap illustrating the expression of T3_3D-specific genes and genes shared with pgMCs_3D or pgADs_3D groups. pgMCs_3D, muscle spheroids formed within 3D suspension culture system. pgADs_3D, adipose-spheroids formed within 3D suspension culture system. T3_3D, spheroids derived from three different cell type sources. **i, j** Overview of essential (EAA), nonessential (NEAA) amino acids, and saturated (SFA), monounsaturated (MUFA), polyunsaturated (PUFA) fatty acids in spheroids and conventional pork. Error bars indicate the means ± SDs, $n = 3$. Values with superscript letters a and b are significantly different across columns ($P < 0.05$).

Similar results were obtained in three independent experiments and represent a significant difference using Two-way ANOVA, followed by Bonferroni's multiple comparisons test. Exact $P$-values are listed in Source Data Fig. 5. **k** GSEA Enrichment Plot of unsaturated fatty acid (UFA) biosynthesis. FDR false discovery rate, ES enrichment score, and NES normalized enrichment score. A one-sided permutation test (10,000 permutations) was used to calculate significance (nominal $p = 9.99 \times 10^{-4}$), with adjustment for multiple comparisons via the false discovery rate (FDR = 0.004). **l** Heatmap illustrating the expression of genes associated with UFA biosynthesis of T3_3D and pig muscle tissue. **m** Appearance of sausages from multiple tissue spheroids, pgEpiSCs, and pork. **n** Textural properties of pork, pgEpiSCs, and pgCM cultured sausages. Error bars indicate the means ± SDs, $n = 3$, ns indicates $p > 0.05$; similar results were obtained in three independent experiments and represent significant differences using One-way ANOVA, followed by Tukey's multiple comparisons test. Exact $P$-values are listed in Source Data Fig. 5. For (**c**), $n = 30$, violin plot cut at lowest and highest value, dotted line: quartile, dashed line: median, dots represent single data points. For (**b**–**g**), similar results were obtained in three independent experiments. Source data are provided as a Source Data file.

medium was replaced with VDM III, and continuous cultivation was carried out. The timing of each differentiation stage has been optimized based on tracing the gene expression profiles across stages. The culture medium configuration is shown in Supplementary Tables 1, 2.

### In vitro tube formation assay
50 μL of Matrigel was added to a 96-well plate and incubated at 37 °C for 30–60 min to allow the gel to solidify. Cells were then plated at a density of $1 \times 10^4$ cells per well onto the Matrigel-coated 96-well plate. After incubation at 37 °C for 4–6 h, imaging was performed on each well using a Reversed Microscope.

### Uptake of Dil-Ac-LDL
Cells were incubated in VDM III containing 20 μg/mL of human Di-Acetylated Low Density Lipoprotein (Human Di-Ac-LDL, YEASEN, 20606ES76) at 37 °C for 4 h. Subsequently, the cells were washed three times with PBS and observed under a fluorescence microscope.

### Alkaline phosphatase (AP) staining
The differentiated cells were fixed with 4% paraformaldehyde (PFA) for 5 min, followed by washing with DPBS (Gibco, Cat# C14190500CP). The AP staining kit (Millipore, Cat# SCR004) was utilized according to the recommended protocol, and then observations were conducted.

### Karyotype analyses
During the active proliferation phase of the cells, the culture medium was replaced with fresh medium with 15% (v/v) KaryoMAX Colcemid Solution (Gibco, 15210-040) for 2 h. The cells were dissociated by TrypLE™ Express (Gibco, Cat No. 12,605,010) into single-cell suspensions, and collected through centrifugation at $1500 \times g$ for 5 min. Subsequently, the cells were suspended in a hypotonic solution of 0.075 M KCl (Sigma, P5405) and incubated at 37 °C for 30 min. Next, the hypotonic cells were fixed by pre-cooled fixative solution (methanol: acetic acid = 3:1) and centrifuged at $1500 \times g$ for 8 min; this process was repeated three times. The resulting cell suspension was then dropped onto pre-cooled glass slides, placed in a 37 °C oven overnight to ensure complete drying, and stained with a Rapid Giemsa Staining kit (BBI Life Science, E6073141) for 15 min before being dried and observed under a microscope.

### Quantitative RT-PCR
Total RNA was extracted using the RNA prep pure Cell/Bacteria Kit (TIANGEN, DP430), and then reverse transcribed into CDNA by Hifair® III 1st Strand cDNA Synthesis SuperMix (YEASEN, Cat# 11141ES60). Soon afterwards, PCR was conducted using 2× RealStar Green Power Mixture (GenStar, A311-05) on a LightCycler 480 II Real Time System

(Roche). The data were analyzed using the comparative CT ($2^{-\Delta\Delta CT}$) method. ΔCT was calculated using *EF1A* as an internal control. Three biological replicates were performed for all experiments. Primer design was carried out using the NCBI primer design tool, which can perform BLAST searches on public databases to ensure the specificity of the primers (https://www.ncbi.nlm.nih.gov/tools/primer-blast). Primer information qRT-PCR was shown in Supplementary Table 3.

### Immunofluorescence staining and imaging
The cells were fixed in 4% PFA solution at room temperature for 30 min. After cleaning with wash buffer (DPBS containing 0.1% Tween 20 (Solarbio, T8220) and 0.01% Triton X-100 (Solarbio, T8200)) three times, 0.05% Triton X-100 was added for 15 min; and rinsing with wash buffer three times again, blocked with 3% BSA (Sigma, Cat# A1470) at room temperature for 1 h. Then the primary antibodies were added for incubation overnight at 4 °C, and washed three times using the wash buffer. Secondary antibodies were incubated at room temperature for 1 h, and washed three times with the same wash buffer. Finally, the nuclei were stained with DAPI (Roche Life Science, 10236276001) for 3 min, which was then observed and photographed under a fluorescence microscope. The antibodies used are listed in Supplementary Table 4.

### Myogenic differentiation of pgEpiSCs by serum-free induction
For myogenic differentiation, pgEpiSCs were dissociated into single cells by Accutase cell dissociation reagent without feeder, then inoculated in a 6-well plate at a density of $2 \times 10^5$ cells/well in feeder-free medium supplemented with 10 μM Y-27632. Single cells of pgEpiSCs were resuspended in MDM I and maintained for 3 days. Cells were harvested using TrypLE after 3 days of induction and inoculated in MDM II. Then the culture was replaced with MDM III for 2 days. After 2 days of induction, MDM IV was supplied and maintained for 4 days. Finally, MDM V was supplied for 20–25 days and replaced with N2 differentiation medium. The culture medium configuration is shown in Supplementary Tables 1, 2.

### 2D monolayer coculture of pgEpiSCs-derived myoblast and preadipocyte
For coculture of pgEpiSCs-myoblast and preadipocyte, a suspension of myoblast and preadipocyte (at a ratio of 0:10, 1:9, 3:7, 5:5, 7:3, 9:1, 10:0) was seeded in twelve-well plates containing proliferation medium (BM (DMEM/F12 supplemented with 1% PS, 1% NEAA, 0.1 mM β-mercaptoethanol, 15% KOSR, and 200 μM ascorbic acid) with 10 ng/ml IGF-1, 10 ng/ml HGF, 10 ng/ml FGF2, and 10 ng/ml hEGF). While cell fusion reached 90%, differentiation was initiated by switching to differentiation medium. The design of the two cultivation programs is presented

as follows: (I) The cells were incubated in differentiation medium (ADM IV: MDM V = 1:1) for 6 days, and in maintenance medium (ADM V: N2 medium = 1:1) for 4 days. (II) The cells were maintained in ADM IV for 6 days, and then maintained in ADM V for 3 days. The cells were maintained in a 5% $CO_2$ humidified incubator at 37 °C, with medium exchanges every 2 days.

## Fabrication of animal-component-free 3D edible plant-based polysaccharide scaffolds

The plant-based 3D edible scaffolds were fabricated via an ion-crosslinking solvent displacement method. In brief, a homogeneous mixture was prepared by combining 3% (w/v) konjac glucomannan (KGM) and 3% (w/v) sodium alginate (SA) solution at a ratio of 1:1, followed by the addition of 2% (w/v) $CaCl_2$ (in 75% ethanol aqueous solution). Then, the mixture was transferred into a 24-well plate (0.5 g/well) and pre-frozen at −20 °C for 12 h. The scaffolds were subsequently obtained by introducing 2% (w/v) $CaCl_2$ aqueous ethanol solution into each well and allowing solvent displacement to proceed for 24 h.

## 3D plant-based edible scaffold cell culture

The cells cultured in 2D were dissociated using TrypLE™ Express (Gibco, Cat No. 12,605,010) and then resuspended in a small volume of culture medium. They were subsequently seeded into 3D edible scaffolds at a density of $1.0 \times 10^6$ cells/mL, allowing for adhesion for 5 h before supplementing with the corresponding culture medium. The pgEpiSCs-preadipocytes were cultured in ADM IV for 6 days, followed by ADM V for an additional 4 days. The 1:1 mixed cell of pgEpiSCs-derived myoblast and preadipocyte were cultivated in differentiation medium (ADM IV: MDM V = 1:1) for 6 days, followed by an additional 4 days in maintenance medium (ADM V: N2 medium = 1:1). All 3D cell cultures were maintained in an incubator at 37 °C with 5% $CO_2$, with media changes performed individually.

## Spheroids formation and cultivation

For the formation of spheroids, a cell density of $5.0 \times 10^5$/mL was utilized with a non-adhesive cell culture dish. Move the dish on the horizontal rotators at 37 °C, 5% $CO_2$, and 100% humidity. The rotation rate is at 70 rpm per min.

## Culture of muscle spheroids

pgMPCs were cultured in 2D and dissociated using TrypLE™ Express at a density of $5.0 \times 10^5$ cells/mL. The culture medium was sequentially replaced as follows: MDM III medium for 2 days, followed by MDM IV medium for 4 days, and finally, MDM V medium was supplied and maintained for 30 days.

## Culture of adipocyte spheroids

pgAPCs were cultured in 2D and dissociated using TrypLE™ Express at a density of $5.0 \times 10^5$ cells/mL. The culture medium was sequentially replaced as follows: ADM III medium for 5 days, followed by ADM IV medium for 6 days, and finally, ADM V medium was supplied and maintained for 30 days.

## Culture of vascular spheroids

pgVPCs were cultured in 2D derived from pgEpiSCs for 6 days and dissociated using TrypLE™ Express at a density of $5.0 \times 10^5$ cells/mL. The culture medium was sequentially replaced as follows: VDM II medium for 2 days, followed by VDM III medium, which was supplied and maintained for 30 days.

## Culture of muscle-adipocyte spheroids

pgMPCs and pgAPCs were cultured in 2D and dissociated using TrypLE™ Express. Cells were mixed at (9:1, 8:2, 7:3, 5:5, 3:7, 2:8, 1:9) ratio and seeded at a density of $5.0 \times 10^5$ cells/mL. The culture medium was

sequentially replaced as follows: MADM I medium for 5 days, followed by MADM II medium for 9 days, and finally, MADM III medium was supplied and maintained for 16 days.

## Culture of muscle-adipose-vascular spheroids

pgMPCs, pgAPCs, and pgVPCs were cultured in 2D and dissociated using TrypLE™ Express. Cells were mixed at (2: 2: 1, 1: 1: 1, 1: 1: 2) ratios and seeded at a density of $5.0 \times 10^5$ cells/mL. The culture medium was sequentially replaced as follows: MAVDM I medium for 5 days, followed by MAVDM II medium for 9 days, and finally, MAVDM III medium was supplied and maintained for 16 days.

## Assay of cell survival efficiency

For cells cultured in 2D adherent conditions and those grown on 3D scaffolds, the cell viability and cytotoxicity were assessed using the Calcein/PI cell activity and cytotoxicity detection kit (Beyotime, C2015M). Cells were double-stained with Calcein-AM (AM) and propidium iodide (PI), followed by observation under a fluorescence microscope, and flow cytometry (BD FACSVerse) was used to quantify the cell viability.

For the spheroids cultured in a 3D suspension, Calcein/PI staining was performed, and the images were taken using a confocal microscope (Nikon, A1 HD25), and flow cytometry was used to quantify cell viability.

## Flow cytometry

Flow cytometric analysis. Cells were digested by TrypLE for 5 min into single cells and centrifuged at $1500 \times g$ for 5 min, which were washed three times with DPBS before being resuspended with diluted fluorochrome and incubated at 37 °C for 30 min. For each sample, $\sim 1 \times 10^6$ viable cells are needed. Following incubation, cells may be analyzed directly by flow cytometry. Calcein-AM fluoresces green (excitation/emission maxima: 494/517 nm), whereas PI emits red fluorescence (excitation/emission maxima: 535/617 nm). All procedures must be conducted under low-light or dark conditions. The unstained cells suspended only in the assay buffer and not exposed to any fluorescent dyes were used to control.

## Fixation and cryosectioning

Prior to fixation, the cell culture medium was discarded, and samples were washed with PBS. Spheroids were fixed by 4% PFA for 1 h after incubation in 30% sucrose solution for 30 min. For cryosectioning, spheroids were embedded in Sakura Tissue-Tek® O.C.T. Compound (Sakura, 4853) within small molds. Subsequently, the bottom of the embedding box was rapidly frozen by liquid nitrogen, and the samples were cooled to −80 °C and sliced into 8 μm of thickness at a chamber temperature (−20 °C) using a cryotome (MEV, SLEE).

## H&E analysis

The tissue sections were washed twice in ddH2O and then stained with haematoxylin (Sigma-Aldrich, MHS16) and eosin (Sigma-Aldrich, HT110116) and observed under a microscope (Leica, DM5500B).

## Scanning electron microscope (SEM) imaging

The scaffolds cultured with differentiated cells and the spheroids were fixed in 4% PFA for 1 h, followed by washing with DPBS for 3 times, and then dehydrated using a series of ethanol solutions at increasing concentrations (30%, 50%, 70%, 80%, 90%, and 100%), each step for 30 min at room temperature. The dehydrated samples were sputter-coated with Pt/Pd on SEM (HITACHI, TM-4000 plus) stubs and subsequently imaged using scanning techniques at an acceleration voltage of 5 kV.

## Confocal microscopy

After cell seeding, the scaffolds were washed with HBSS (Beyotime, C0219) and fixed at room temperature using 4% PFA for 1 h. The

staining method on 3D was identical to that of immunofluorescence staining; following staining, the samples were rinsed with HBSS. Images were captured using a laser scanning confocal microscope (Nikon, A1 HD25).

## Transcriptome sequencing

PolyA mRNA was enriched from total RNA using Oligo (dT) magnetic beads and fragmented to ~300 bp by ion shearing. First- and second-strand cDNA were synthesized using random hexamer primers and reverse transcriptase. Libraries were constructed, amplified by PCR, and size-selected for ~450 bp fragments. Library quality and concentration were assessed with an Agilent 2100 Bioanalyzer. Indexed libraries were pooled, diluted to 2 nM, denatured, and sequenced on the Illumina platform using paired-end sequencing.

## RNA-seq data processing

Raw sequencing reads were quality-checked using FastQC (v0.11.9) and trimmed to remove adapters and low-quality bases using Trimmomatic (v0.39) with default parameters. Processed reads were aligned to the pig reference transcriptome (Sus scrofa, Ensembl release 113, cDNA sequences: Sus_scrofa.Sscrofa11.1.cdna.all.fa.gz, downloaded from https://ftp.ensembl.org/pub/release-113/fasta/sus_scrofa/cdna/) using Kallisto (v0.46.0)[55]. Gene-level read counts and Transcripts Per Million (TPM) values were summarized using the associated gene annotation file (*Sus scrofa*, Ensembl release 113, GTF: Sus_scrofa.Sscrofa11.1.113.gtf.gz, downloaded from https://ftp.ensembl.org/pub/release-113/gtf/sus_scrofa/) via the tximport package (v1.18.0) in R (v4.3.0).

## Principal component analysis

To explore sample variability and assess data quality, principal component analysis (PCA) was performed on variance-stabilized read counts generated by DESeq2 (v1.30.0). The top genes with the highest variance across samples were selected for PCA computation. Results were visualized using ggplot2 (v3.4.0) in R.

## Differential gene expression analysis

Differential gene expression analysis was conducted using DESeq2 (v1.30.0)[56]. The Wald test was applied to identify differentially expressed genes (DEGs) between different sample groups. P-values were adjusted for multiple testing using the Benjamini–Hochberg false discovery rate (FDR) method. Genes with FDR < 0.05 and |log2 fold change| ≥ 1 were considered significantly differentially expressed. A heatmap of DEGs was generated using pheatmap (v1.0.12) in R to visualize expression patterns across samples.

## Gene functional enrichment analysis

Functional enrichment analysis of DEGs was performed using Metascape[57] with the Gene Ontology Biological Process (GO-BP) database. Terms with a minimum gene count ≥3, adjusted *P*-value < 0.01, and enrichment factor ≥1.5 were considered significant. Related terms were grouped into clusters based on similarity. Key enriched pathways and terms were visualized using bar plots generated in R.

## Gene Set Enrichment Analysis (GSEA)

GSEA was performed on the ranked list of genes using the fgsea R package (v1.26.0) to identify signaling pathways that were significantly activated in cell samples relative to native meat samples. Visualization of the GSEA results was implemented using the ggplot2 and pheatmap packages in R.

## Amino acid composition analysis

50 mg of the solid sample were accurately weighed and transferred into a hydrolysis tube, with 10 mL of a 1:1 mixture of analytical grade hydrochloric acid (approximately 6 M). The tube containing the sample was tightly sealed after being purged with nitrogen gas for 30 s to eliminate oxygen. Next, the sealed tube was placed in an oil bath and subjected to hydrolysis at a temperature of 110 °C for a duration of 22–24 h. After the tube was cooled to room temperature, the hydrolysis products were filtered through a 0.45 μm membrane filter into a 50 ml volumetric flask. 2 mL of the diluted samples was transferred to a rotary evaporator and evaporated at 45 °C until the samples approached dryness, leaving a small amount of solid residue or trace moisture at the bottom of the flask. Subsequently, 2 mL of sample buffer was added to the flask to completely dissolve the residue. After filtration through a 0.45 μm filter, the sample is ready for analysis using an amino acid analyzer (LA8080, Hitachi, Tokyo, Japan).

## Fatty acid composition analysis

A sample weighing 30–50 mg was accurately measured and dissolved in ethanol, then diluted to a final volume of 25 mL. An aliquot of 1 mL from the prepared solution was transferred into a 15 mL centrifuge tube, followed by the addition of 2 mL of a 5% hydrochloric acid-methanol solution, 3 mL of a chloroform-methanol mixture (1:1, v/v), and 100 μL of methyl nonadecanoate (internal standard). The centrifuge tube was then incubated in a water bath at 85 °C for 1 h. After the reaction, the test tube was allowed to cool to room temperature. Subsequently, 1 mL of n-hexane was added to the test tube and vortexed for 2 min to extract the target compounds. The mixture was allowed to stand for 1 h for phase separation. From this mixture, an upper organic layer (n-hexane phase) measuring 100 μL was taken and diluted with n-hexane up to a total volume of 1 mL. After filtration through a 0.45 μm filter, the sample was prepared for analysis.

This method utilizes a TG-5MS (30 m × 0.25 mm × 0.25 μm) capillary column for separation, with a programmed temperature gradient to optimize the elution of analytes. Hold at 80 °C for 1 min. ramp at 10 °C min$^{-1}$ to 200 °C, then ramp at 5 °C min$^{-1}$ to 250 °C, and finally ramp at 2 °C min$^{-1}$ to 270 °C, hold for 3 min. The Gas Chromatograph is coupled to a mass spectrometer operating in electron ionization (EI) mode, with specific temperature (270 °C) settings for the ion source and transfer line. The scan range of 30–400 amu ensures comprehensive detection of target compounds, while a solvent delay of 5 min prevents detector saturation from the solvent peak.

The content of each fatty acid in the sample is calculated according to the formula:

$$X_i = \frac{A_{si} \times m_{skli} \times F_j}{A_{stdi} \times m} \tag{1}$$

Where:

$X_i$: Content of each fatty acid in the sample, expressed in milligrams per kilogram (mg kg$^{-1}$).

$A_{si}$: Peak area of each fatty acid in the sample test solution.

$m_{skli}$: Mass of the standard (fatty acid triglyceride) in the standard working solution used for preparing the standard test solution, expressed in milligrams (mg).

$F_j$: Conversion factor for transforming fatty acid triglycerides into fatty acids.

$A_{stdi}$: Peak area of each fatty acid in the standard test solution.

$m$: Mass of the sample weighed, expressed in kilograms (kg).

## GC-MS analysis

Agilent 7890B-7000C series of GC-MS instruments (Agilent Technologies Inc., CA, USA) was used in this method. Two groups were set up: the control group with pork and the experimental group with pgCM. Transfer 1 g of the sample into a 20 mL extraction vial and add 10 μL of a 50 μg/mL solution of 2-methyl-3-heptanone (internal standard). Quickly seal the vial to prevent loss of volatile compounds. Condition the SPME fiber (50/30 μm PDMS/CAR/DVB, 2 cm; Supelco Analytical, Bellefonte,

Pennsylvania, USA) in the GC-MS injection port at 250 °C until no contaminant peaks are detected. Place the sealed sample vial on the SPME apparatus and set it to 60 °C. Insert the SPME fiber through the vial septum into the headspace, positioning it approximately 1.0 cm above the sample surface. Expose the fiber for 30 min to allow adsorption of volatile compounds. After extraction, retract and remove the fiber from the vial, then immediately insert it into the GC-MS injection port. Expose at 250 °C for 3 min to desorb analytes.

This method uses a DB-5MS (30 m × 250 μm, 0.25 μm) capillary column with an optimized temperature gradient for target compound elution: hold at 40 °C for 1 min, ramp at 6 °C min$^{-1}$ to reach 100 °C, then ramp at 10 °C min$^{-1}$ to 230 °C and hold for 3 min. The GC is coupled with a mass spectrometer operating in electron ionization (EI) mode; specific temperatures are set for the ion source (230 °C), transfer line (230 °C), and quadrupole (150 °C). The mass scan range is set from 40 to 550 m/z for comprehensive analyte detection. Qualitative analysis utilizes NIST17 library data and retention times, and the volatile compounds were characterized on the basis of matching scores and in light of the literature; quantitative analysis relies on peak area normalization from total ion chromatograms. The trial data organization and statistics were conducted using Excel 2021. Each sample was tested three times to ensure accuracy.

### Texture profile analysis

TPA was measured by a texture analyzer (TA.XTplus, Stable Micro Systems Ltd (TPA mode, P/75 probe)). The cylindrical sausage samples, with a diameter of 15 mm and a height of 20 mm, were positioned at the center of the test platform, followed by TPA analysis. The parameters were set as follows: the pre-test speed was 1 mm s$^{-1}$, the test speed was 1 mm s$^{-1}$, and the post-test speed was 1 mm s$^{-1}$; trigger force set to 0.005 kg, trigger mode automatic; the degree of pressing down is 50%, the interval time is 5 s, and the reciprocation is carried out 2 times.

### The production of cultured and farmed pork sausage

Processing steps: Mincing → Ingredient weighing → Chopping/mixing → Filling/casing → Cooking → Cooling

Cooking conditions: temperature at 82 °C, 45 min, casing diameter 15 mm (using 15 mL centrifuge tubes as an alternative). The recipe for sausages is shown in Supplementary Table 5.

### Statistical analysis

All values in the graphs are reported as the mean ± SD. GraphPad Prism (v10) was used to construct diagrams. One-way ANOVA, followed by Dunnett's multiple comparisons test, was used to analyze data of qRT-PCR in Fig. 1c. One-way ANOVA, followed by Tukey's multiple comparisons test, was used in Fig. 5n. Two-tailed Student's t-test was used to analyze data in Figs. 1e, f, 3n, and Supplementary Figs. 2d, 4g, 5b, d, p, 7c, 9j, 10h. Two-way ANOVA, followed by Dunnett's multiple comparisons test, was used to analyze data of qRT-PCR in Fig. 2c–e, 3f, g and Supplementary Figs. 1b, c, e, f, 3b, c, 5i, j; Two-way ANOVA, followed by Bonferroni's multiple comparisons test, was used in Fig. 5i, j. Supplementary Figs. 8c, 10c, d–g. P-values less than 0.05 were considered statistically significant. Exact P-values are listed in Source Data. Figure legends indicate the number of independent experiments performed in each analysis. Each experiment had been repeated for reproducibility.

### Image processing and analysis

All schematic illustrations were produced by Adobe Illustrator CC 2019. ImageJ (v1.8.0) was used for quantification of the fluorescent images, and GraphPad Prism (v10) was used for statistical analysis and the creation of statistical graphs.

### Reporting summary

Further information on research design is available in the Nature Portfolio Reporting Summary linked to this article.

### Data availability

The raw RNA-seq data generated in this paper have been deposited in the Genome Sequence Archive (GSA) at the National Genomics Data Center, China National Center for Bioinformation (NGDC/CNCB), under accession number CRA026288, and are publicly accessible at https://ngdc.cncb.ac.cn/gsa/browse/CRA026288. The data generated in this study are provided in the Supplementary Information/Source Data file. Source data are provided with this paper.

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

## Acknowledgements

This work was supported by the Biological Breeding-National Science and Technology Major Project (2023ZD0407503 to J.H.), the Natural Science Foundation of China (32370846 to J.H., 31970825 to J.H., 32402757 to M.Z.), the National Key R&D Program of China (2022YFD1302201 to J.H.), the fellowship of China Postdoctoral Science Foundation (2022M720168 to M.Z.), China National Postdoctoral Program for Innovative Talents (BX20220344 to M.Z.), Pinduoduo-China Agricultural University Research Fund (PC2024A01001 to J.H.), the 2115 Talent Development Program of China Agricultural University and High-performance Computing Platform of China Agricultural University.

## Author contributions

J.H. conceptualized this project. J.H., S.C., and A.M. supervised the overall experiments. Y.Y. and G.Z. performed pgEpiSCs culture in vitro and characteristics analysis of differentiated cells. J.H., Y.Y., R.L., and

M.Z. coordinated and performed bioinformatics analysis of the RNA-seq sequencing data. Y.Y., G.Z., M.Z., and T.W. performed spheroid culture in vitro and characteristics analysis. J.H., Y.Y., A.M, Y.Z., and Y.C. performed preparation of cell-based meat, nutritional components, and texture analysis. Y.Y., G.Z., L.L., and X.F. completed the preparation, cell inoculation of 3D edible scaffolds, and SEM. M.Z. performed pgEpiSCs derivation. Y.Y., G.Z., T.W., and L.Q. performed flow cytometric analysis. Y.Y., G.Z., Y.W., S.W., and H.Z. performed qRT-PCR primer design, quantitative validation, and immunofluorescence image processing. Y.Y., X.C., and M.Z. performed the preparation of the experimental schematic. J.H. and Y.Y. performed manuscript writing, review, and editing.

## Competing interests

The authors declare no competing interests.
