## [Transparent Peer Review file · Nature Communications]

Generation of multitissue cell-cultivated meat via multidirectional differentiation of stable porcine epiblast stem cells

Corresponding Author: Professor Jianyong Han

Version 0:

Reviewer comments:

Reviewer #1

(Remarks to the Author)

The manuscript presents an innovative approach using directed differentiation of porcine epiblast stem cells (pgEpiSCs) into muscle, adipose, and vascular endothelial cells under serum- and animal component-free conditions. This strategy enables the creation of cultivated meat with structural and nutritional complexity that mimics traditional pork. The concept of using a single stem cell source to generate multi-tissue constructs is timely and impactful in the field of cellular agriculture.

However, the manuscript lacks sufficient clarity in describing the mechanistic rationale and biological insight behind the protocol design. Moreover, several aspects of novelty, experimental design, and interpretation require more rigorous elaboration before the work can be considered for publication.

Major Points to Address

1. Mechanistic Justification of Protocol Design:

The manuscript should elaborate on how the differentiation protocols were developed. What prior biological knowledge or hypotheses guided the choice of specific factors (e.g., CHIR99021, SB431542, LDN193189, VEGF)? The treatment order, duration, and rationale for specific combinations in Figure 1a must be clearly explained. Currently, the approach appears as a collection of component lists without mechanistic reasoning.

2. Clarify Novelty and Differentiation from Prior Work:

The authors have previously published similar strategies using pgEpiSCs and the same inhibitors. While this work introduces additional lineages, the conceptual advancement is unclear. The authors must explicitly describe what is novel compared to their previous publication (Nature Communications 14, 8163 (2023)). What key innovation or technical advancement does this manuscript bring?

3. Generalizability of Protocol:

Discuss whether the directed induction protocols are specific to pgEpiSCs or potentially applicable to other PSC types. This would increase the relevance of the work beyond the porcine model.

4. Scaffold and 3D Culture System:

The authors mention scaffold-based and scaffold-free 3D culture systems but do not adequately describe the scaffold materials or their design. Although previous work is cited, a brief but clear summary is needed in this manuscript to ensure readers can interpret the experimental context independently.

5. Mechanism of Multilineage Spheroid Assembly:

The authors should explain whether the spatial organization of cells in the spheroids arises from random aggregation or specific cell–cell and cell–matrix interactions. Additionally, they must address whether the final tissue composition (e.g., muscle-to-fat ratio) is controllable through seeding ratios, and if so, provide supporting quantitative data.

6. Evaluation of Nutritional Content:

The comparison of amino acid and fatty acid profiles between cultivated meat (CM) and traditional pork lacks sufficient normalization. The manuscript should provide absolute total amino acid and fatty acid contents per unit mass. Current figures (e.g., 5i, 5j) only show relative proportions within each group, which does not fully support the claim of nutritional similarity.

7. Texture and Flavor Similarity Claims:

The similarity in mechanical texture and flavor profiles between CM and pork requires more substantial explanation. What are the proportions of muscle, fat, and ECM in both CM and pork? If they are significantly different, how is the texture similarity achieved? The current explanation that collagen and ECM contribute to texture must be supported with quantitative data.

8. PUFA/SFA Ratio Difference:

The PUFA/SFA ratio of CM is reported to be higher than that of traditional pork. The manuscript should explain this discrepancy, including whether it results from cell type composition, media composition, or other culture conditions.

9. Scalability and Industrial Relevance:

The authors need to strengthen the discussion regarding how their strategy improves scalability and reproducibility of cultured meat. What are the implications for industrial applications, especially under suspension culture conditions?

10. Literature Context and Claims:

The authors claim that most previous research on cultured meat focused on muscle tissue, but this is not entirely accurate. Numerous studies have explored fat and other supporting tissues. The discussion section should acknowledge and properly cite relevant literature, such as:

- Communications Biology 5, 927 (2022)
- Materials Today Bio 21, 100720 (2023)
- Food Chemistry 460, 140696 (2024)
- Nature Food 4, 35–50 (2023)
- Nature Communications 12, 5059 (2021)

11. Typographical Corrections:

In line 207 of the main text, the citation “Fig. g” should be corrected to “Fig. 5g.”

The manuscript proposes a promising strategy for constructing multitissue cultivated meat. However, the lack of mechanistic depth, unclear novelty, insufficient data normalization, and weak discussion limit its current impact. A thorough revision addressing the points above is required to meet the standards of Nature Communications.

Reviewer #2

(Remarks to the Author)

The paper describes the generation of multitissue cell-cultivated meat using stable porcine epiblast stem cells. Researchers achieved multidirectional differentiation of these stem cells to produce various tissue types relevant for cultivated meat production. The authors previously already showed the differentiation of muscle tissue from pig epiblast stem cells (Generation of three-dimensional meat-like tissue from stable pig epiblast stem cells | Nature Communications) and extend this line of research here with a multi-cell type approach.

The presented paper claims to have developed a multi-tissue cultured meat product using porcine epiblast stem cells and 3D suspension culture for the application in cultured meat.

The publication contains several interesting routes of investigation that are relevant for the field of cultured meat and offer a good starting point for further research. The authors claimed to have developed a serum- and animal component free differentiation system that enables direct induction of muscle, adipose, and vascular endothelial cells.

The main body of evidence the authors present is done in 2D, using also animal-derived components, and only limited evidence is presented for the 3D culture system (see the detailed feedback under “Major points”).

Major points

Claim 1: Serum- and animal component free differentiation system

Several initial steps in the development to derive muscle and adipogenic progenitors used animal derived components (like feeder cells for expanding piEpiSCs and gelatin for pre-adipocyte differentiation).

While showing a completely animal-free process may be out of scope of this paper, those are relevant steps in the development of cultured meat and this limitation in the presented protocol has to be discussed and an outlook has to be given

Claim 2: Direct induction to muscle, adipose and vascular endothelial cells

In 2D, the authors show indeed directed differentiation from EpiSCs towards the 3 different tissues. There are, at least for adipogenic cells, similarities to in vitro derived adipocytes (from differentiated FAPs), but the EpiSC-derived adipocytes show lower maturity though in regards to the gene expression.

More importantly, little is shown about the size and maturity of the fat droplets in those cells. The fluorescent images in Fig. 1 are relatively small and it is hard to see if unilocular droplets are present. Here, better pictures including a FAP-derived control are advised to enable the reader to judge the differentiation efficiency better

Claim 3: Formation of tissue mimetic spheroids

While some data is shown on culturing adipogenic and myogenic progenitors in 3D (both alone and in co-culture), the data is missing more proof to be seen as a convincing model for a scalable culture:

Authors showed 30 days of culture with a final spheroid size of > 1mm and it is not believable that there was no cell death, e.g. in the form of a dead core, present. Just showing pictures from the surface of the structures is in this case not enough, cryosections would be needed to judge the inner core of structures for viability

While accumulation of fat droplets in the structures indicates successful adipogenic differentiation, the staining for myogenic

differentiation is not convincing: maturing skeletal differentiation leads to more alignment and cellular fusion, neither is shown in the presented data.

The co-culture (Figure 5) shows very limited differentiation, especially towards muscle. It doesn't seem convincing that all initially included cell types actually survived in the 3D structure or if one cell type may have overgrown the others. Again, while some marker expression is shown, no mature muscle (with alignment and fusion) is shown in those structures. It seems unlikely that those structures have a very different protein composition to undifferentiated material

Abstract and introduction make it sound as if the entire process could be done in 3D and in co-culture, albeit Fig. 5 indicates that only the last maturation step is done in 3D co-culture

Claim 4: Replicates textural and nutritional value of conventional produced pork

It is interesting and promising to see the similarities between conventional pork, but as detailed above the limited differentiation efficiency shown in the 3D structures makes it hard to believe that the impact of the differentiation process is relevant. Here, an undifferentiated control cell mass would be needed to prove that the presented differentiation process actually makes an impact. It is possible that the hardness and guminess are more influenced by the production method itself. Here some more detail could help as well.

Also: while it is great to see the analytical specifications of their cultured meat product, it does not seem that actual tastings have taken place. Claims in regards to the taste therefore should be taken carefully .

Minor points

Fig. 1a typos (SB421542 instead of SB431542) and Roaiglitazone instead of Rosiglitazone

Fig 3a: A more detailed description of the final protocol for the co-culture (including details on the used basal medium and potential reasoning for the used growth factors) would be helpful. Generally, the method description is very superficial and does not enable replication. Since this is the most important novelty in the present publication, more detail would be expected

Extended Fig 4: What does the edible scaffold consist of?

Also not sure if the culture in the scaffold adds anything to the presented story, as authors mention themselves that 2D expansion and scaffolds are not a viable option for the future

Discussion: The discussion missed an evaluation of the current limitation of their approach. As said, this is a promising starting point for cultured meat research, but honest discussion of limitations (like that initial steps are done in 2D and the use of very expensive and not food safe materials) should be discussed and an outlook should be given IF the authors have reason to believe that those limitations can be overcome.

Reviewer #3

(Remarks to the Author)

Version 1:

Reviewer comments:

Reviewer #1

(Remarks to the Author)

Overall, I find that the authors have addressed the major concerns raised in the initial review process in a thorough and convincing manner. The revised manuscript represents a substantial improvement in terms of mechanistic clarity, experimental rigor, and balance of claims, and now reaches a level of maturity that is largely appropriate for Nature Communications.

Reviewer #2

(Remarks to the Author)

The authors have addressed all concerns extensively and satisfyingly

Reviewer #3

(Remarks to the Author)

RE: “Generation of multitissue cell-cultivated meat via multidirectional differentiation of stable porcine epiblast stem cells” (NCOMMS-25-46555-T)

Point-by-point response to reviewers’ comments:

REVIEWER COMMENTS

Reviewer #1 (Remarks to the Author):

The manuscript presents an innovative approach using directed differentiation of porcine epiblast stem cells (pgEpiSCs) into muscle, adipose, and vascular endothelial cells under serum- and animal component-free conditions. This strategy enables the creation of cultivated meat with structural and nutritional complexity that mimics traditional pork. The concept of using a single stem cell source to generate multi-tissue constructs is timely and impactful in the field of cellular agriculture.

Response: Thank you for the positive comments.

However, the manuscript lacks sufficient clarity in describing the mechanistic rationale and biological insight behind the protocol design. Moreover, several aspects of novelty, experimental design, and interpretation require more rigorous elaboration before the work can be considered for publication.

Response: Thanks for your comment and suggestion. We have added additional experiments and results to the response, hoping to address the concerns.

Major Points to Address

1. Mechanistic Justification of Protocol Design:

The manuscript should elaborate on how the differentiation protocols were developed. What prior biological knowledge or hypotheses guided the choice of specific factors (e.g., CHIR99021, SB431542, LDN193189, VEGF)? The treatment order, duration, and rationale for specific combinations in Figure 1a must be clearly explained. Currently, the approach appears as a collection of component lists without mechanistic reasoning.

Response: We sincerely appreciate your insightful comments and suggestion. We apologize for any confusion that may have arisen due to the ambiguity in our wording. In terms of differentiation protocols, we selected a combination of small molecules and optimized the temporal scheduling based on tracing the gene expression profiles across stages according to the biological characteristics of porcine stem cells. We have revised the original manuscript and clarified the roles of factors.

For adipocyte differentiation, we employed a stage-specific directed differentiation strategy, first guiding pgEpiSCs toward mesenchymal-like cells (adipose progenitor cells), followed by adipogenic induction. The basal medium used was the previously established pgEpiSC differentiation medium (BM) developed by our research group¹.

To initiate mesoderm differentiation, the factors selected (CHIR99021, SB431542) are well-characterized regulators of developmental signaling pathways, and their choice was based on extensive prior knowledge of mesodermal lineage specification^{1, 2, 3}. Pluripotent genes (*OCT4*, *SOX2*, *NANOG*) and lineage-specific markers of the three germ layers (*T*, *GATA6*, *PAX6*) were employed as molecular indicators. Following 3 days of induction, the mesodermal marker *T* exhibited the highest expression level, while pluripotency-associated genes were significantly downregulated. These findings indicate that day 3 represents the optimal time point for completion of the first differentiation stage.

To initiate mesenchymal-like cells (adipose progenitor cells) differentiation, CHIR99021, LDN193189, FGF2, EGF were added to promote mesodermal lineage specialization and epithelial-mesenchymal transition (EMT)^{1, 4, 5}. Adipocyte progenitor cell markers—including *DLK1*, *PDGFRA*, *Zfp423*, and *VIM*—were monitored to assess cellular progression. By tracking the gene expression in a sequential manner, we determined that day 12 represented the optimal time point for differentiation. An additional 6 days were required for the maintenance of APC-stage cells and subsequent cell expansion and proliferation. pgEpiSC-derived APCs (pgAPCs) exhibit MSC characteristics, as evidenced by the expression of the *PDGFRA*⁺*CD29*⁺*CD45*⁻ and the genes *ITGB1*, *Thy1*, *ENG*, notably, they lack expression of *PECAM1*, and *PTPRC*, which is consistent with the findings of *in vivo* experimental findings.

To initiate adipocyte-like cells differentiation, we induced adipogenesis via the classical cocktail approach, with additional supplementation of ITS (insulin-transferrin-selenium) and the

PPAR γ agonist rosiglitazone in serum-free culture. The optimal differentiation time 6 days were determined by detecting the expression of adipogenesis-related genes. And upon further maintenance culture, Oil Red O, BODIPY, and Nile red staining confirmed robust lipid droplet (LD) accumulation in pgEpiSC-derived adipocytes (pgADs), with positive expression of the adipocyte maturation marker.

For endothelial cell differentiation, the differentiation protocol is relatively mature and exhibits high interspecies conservation^{6, 7}. For the basal medium, we employed the basal pgEpiSCs differentiation medium (BM) previously published by our research group¹. We adapted published protocols by supplementing the basal medium for pgEpiSC differentiation with small molecules involved in vascular lineage specification, and further optimized the differentiation duration for each stage by RT-PCR of genes expression.

These above have been thoroughly elaborated in the added result and method of revised manuscript (**as shown in Extended Data Fig. 1e, f and Lines 70-73, 81-91, 346-356, 377-383**).

References:

- 1 Zhu, G. *et al.* Generation of three-dimensional meat-like tissue from stable pig epiblast stem cells. *Nat Commun* **14**, 8163 (2023).
- 2 Chen, Y. S. *et al.* Small molecule mesengenic induction of human induced pluripotent stem cells to generate mesenchymal stem/stromal cells. *Stem Cells Transl Med* **1**, 83-95 (2012).
- 3 Kempf, H. *et al.* Bulk cell density and Wnt/TGF β signalling regulate mesendodermal patterning of human pluripotent stem cells. *Nat Commun* **7**, 13602 (2016).
- 4 Lian Q, *et al.* Functional mesenchymal stem cells derived from human induced pluripotent stem cells attenuate limb ischemia in mice. *Circulation*. 9;121(9):1113-23 (2010).
- 5 Lamouille, S., Xu, J. & Derynck, R. Molecular mechanisms of epithelial-mesenchymal transition. *Nat Rev Mol Cell Biol* **15**, 178-196 (2014).
- 6 Orlova, V. V. *et al.* Generation, expansion and functional analysis of endothelial cells and pericytes derived from human pluripotent stem cells. *Nat Protoc* **9**, 1514-1531 (2014).
- 7 Wimmer, R. A., Leopoldi, A., Aichinger, M., Kerjaschki, D. & Penninger, J. M. Generation of blood vessel organoids from human pluripotent stem cells. *Nat Protoc* **14**, 3082-3100 (2019).

2. Clarify Novelty and Differentiation from Prior Work:

The authors have previously published similar strategies using pgEpiSCs and the same inhibitors. While this work introduces additional lineages, the conceptual advancement is unclear. The authors must explicitly describe what is novel compared to their previous

publication (Nature Communications 14, 8163 (2023)). What key innovation or technical advancement does this manuscript bring?

Response: Thanks for your comment and suggestion. At the same time, we appreciate your attention to our previous work. Our prior research focused on establishing a technical system for the directed *in vitro* differentiation of pgEpiSCs into muscle cells under serum-free conditions, integrated with a 3D scaffold free of animal-derived components, thereby enabling the production of cultured meat from porcine pluripotent stem cell-derived lineages.

First of all, in comparison to our prior research, the present study expanded the lineage differentiation potential of pgEpiSCs, achieving successful induction of adipogenic and vascular endothelial cell differentiation. Notably, there have been no previous reports on the adipogenic differentiation of porcine pluripotent stem cells.

Concurrently, we explored various types of cell co-culture approaches. By co-culturing the progenitor cells of these three cell types in a 3D suspension condition, we observed spontaneous self-aggregation, proliferation, and differentiation, thereby constructing multitissue spheroids and thus the preparation of multitissue CM. The entire differentiation process was serum-free, animal-component-free, and no exogenous scaffolds were introduced. This method is expected to offer greater suitability for large-scale bioreactors and industrial production.

In addition, compared with the previous single muscle tissue, the final tissue composition we obtained could be modulated by adjusting the cell seeding ratio, thereby controlling its nutritional components. Moreover, the final CM products are more similar to the texture characteristics and nutritional composition of native pork.

Above all, the novelty of this study lies in the development of multi-tissue cell-cultured meat (CM) via the multi-lineage differentiation of porcine pre-gastrulation epiblast stem cells (pgEpiSCs), enabling the creation of the multi-tissue and nutritionally customized animal CM. These findings could provide a new avenue for the development and future industrialization of multitissue CM, representing a further advancement compared to previous work.

3. Generalizability of Protocol:

Discuss whether the directed induction protocols are specific to pgEpiSCs or potentially applicable to other PSC types. This would increase the relevance of the work beyond the porcine model.

Response: We sincerely appreciate the insightful comment regarding the generalizability of our directed differentiation protocol. According to your suggestion, we performed experiments to see whether our directed differentiation protocols are applicable to other species. Under serum-free conditions, the methods described in this study were employed to induce the differentiation of sheep (sfPSCs) and bovine PSC (bEpiSCs) into muscle cells (MCs), adipocytes (ADs), and endothelial cells (ECs), yielding the following preliminary findings:

- Myogenic Lineage: Differentiated sfPSC-derived muscle cells (sfPSCs-MCs) display a characteristic fibrous morphology and undergo cell fusion. And the formation of multinucleated myotubes expressing mature skeletal muscle markers, including myosin and F-actin (**Response Figure 1a-c**). The myogenic differentiation potential of bEpiSCs has been previously documented².

- Adipogenic Lineage: Distinctive intracellular lipid droplets (LDs) were visualized via Oil Red O staining, confirming terminal adipocyte differentiation. Both sfPSC-derived adipocyte (sfPSC-ADs) and bEpiSC-derived adipocyte (bEpiSC-ADs) showed positive staining (**Response Figure 1d-g**), indicating substantial accumulation of LDs.

- Endothelial lineage: Differentiated sfPSC-derived ECs (sfPSCs-ECs) and bEpiSC-derived ECs (bEpiSCs-ECs) exhibited a flat, polygonal morphology and expressed the EC marker CD31 (**Response Figure 1h-k**).

These results demonstrate that, despite minor variations in cellular morphology across species, the signaling pathways regulating mesodermal fate specification are evolutionarily conserved among these large mammalian species.

Indeed, the establishment of stable porcine pre-gastrulation epiblast stem cells (pgEpiSCs) represents an important event in livestock biotechnology¹. However, we agree that demonstrating the robustness of our protocol across different species is crucial for broadening its relevance. Our findings align with recent studies indicating that pgEpiSCs from pig, bovine, and sheep share a conserved ground-state pluripotency network^{2,3}. Specifically, the successful

derivation and maintenance of these cells using the 3i/LAF culture system or the optimized alternative media developed on its basis suggest a high degree of functional conservation in the early epiblast stage across these species. This shared regulatory environment likely underpins the responsiveness of these cells to similar extrinsic differentiation cues.

So far, our protocol has demonstrated initial evidence of generalizability. Ongoing work by other team members is further exploring its applicability across species, and additional studies are required to fully establish its broad utility.

Response Figure 1: Characterization of sfPSCs-MCs, sfPSCs-ADs, bEpiSCs -ADs, sfPSCs-ECs, and bEpiSCs -ECs.

a, Cellular morphological observation of the differentiation induction process of sfPSCs-MCs. **b, c**, Immunostaining of Myosin and F-actin in sfPSCs-MCs. **d**, Morphological observation of adipogenic differentiation induction process of sfPSCs. **e**, Oil red O staining of sfPSCs-ADs. **f**, Cellular morphological observation of adipogenic differentiation induction process of bEpiSCs-ADs. **g**, Oil red O staining of bEpiSCs-ADs. **h**, Morphological observation of endothelial

differentiation induction process of sfPSCs. **i**, Immunostaining of CD31 in sfPSCs-ECs. **j**, Morphological observation of endothelial differentiation induction process of bEpiSCs. **k**, Immunostaining of CD31 in bEpiSCs-ECs. For all figures, scale bar, 50 μm .

References:

1. Zhi, M. *et al.* Generation and characterization of stable pig pregastrulation epiblast stem cell lines. *Cell Res* **32**, 383-400 (2022).
2. Zhang, J. Y. *et al.* Tracing and Capturing the Epiblast Pluripotency of Sheep Preimplantation Embryos. *Adv Sci* (2025).
3. Zhi, M. L. *et al.* Elucidation of the pluripotent potential of bovine embryonic lineages facilitates the establishment of formative stem cell lines. *Cellular and Molecular Life Sciences* **81** (2024).

4. Scaffold and 3D Culture System:

The authors mention scaffold-based and scaffold-free 3D culture systems but do not adequately describe the scaffold materials or their design. Although previous work is cited, a brief but clear summary is needed in this manuscript to ensure readers can interpret the experimental context independently.

Response: Thanks for your suggestion. We sincerely apologize for the oversight in describing this section. For the fabrication of the 3D scaffold, we employed the method detailed in our previously published work. The scaffold materials and design are described in detail below:

Fabrication of Animal-Component-Free 3D Edible Plant-Based Polysaccharide Scaffolds

The plant-based 3D edible scaffolds were fabricated via an ion-crosslinking solvent displacement method as previously described¹. In brief, a homogeneous mixture was prepared by combining 3% (w/v) konjac glucomannan (KGM), and 3% (w/v) sodium alginate (SA) solution at a ratio of 1: 1, followed by the addition of 2% (w/v) CaCl_2 (in 75% ethanol aqueous solution). Then, the mixture was transferred into a 24-well plate (0.5 g/well) and pre-frozen at $-20\text{ }^\circ\text{C}$ for 12 h. The scaffolds were subsequently obtained by introducing 2% (w/v) CaCl_2 aqueous ethanol solution into each well and allowing solvent displacement to proceed for 24 h.

These above have been thoroughly elaborated in the methods of the revised manuscript **(as shown in Lines 435-441)**.

5. Mechanism of Multilineage Spheroid Assembly:

The authors should explain whether the spatial organization of cells in the spheroids arises from random aggregation or specific cell–cell and cell–matrix interactions.

Response: Thanks for your insightful comment and question. Based on your suggestion, we conducted experiments to investigate the mechanism underlying multilineage spheroid assembly. To elucidate the origin of cellular spatial organization, we established fluorescent protein-labeled reporter cell lines and employed a combined approach of directed differentiation and self-aggregation assays.

Specifically, we established the pgEpiSCs-mCherry fluorescent reporter cell line (expressing red fluorescence) and validated its adipogenic differentiation potential using an identical protocol as described in this paper. The differentiated cells exhibited adipocyte-specific characteristics and showed positive BODIPY staining, indicating abundant LDs accumulation (**Response Figure 2a**). Additionally, a pgEpiSCs-NLS-GFP cell line (green fluorescence exhibiting nuclear localization), which expresses nuclear-localized green fluorescence was established and directionally induced to differentiate into myocytes, thereby validating its myogenic differentiation capacity. The derived muscle cells (pgMCs) were found to express Myosin protein (**Response Figure 2b**). Subsequently, to elucidate the mechanism underlying spheroids formation, we performed mixed suspension cultures for self-aggregation assays, using two combinations of cells: (1) pgMPCs-GFP and pgAPCs-WT cells (**Response Figure 2c**), and (2) pgMPCs-GFP, pgAPCs-mCherry, and pgVPCs-WT cells (**Response Figure 2d**). These cells marked with different colors could combine with each other to form spheroids. Although the different types of cells were not evenly distributed, we tend to favor the conclusion of random aggregation (**Response Figure 2e, f**). These results suggest that the cells are capable of aggregating and proliferating, and forming structures similar to muscle, fat and endothelium, although they still cannot be compared with normal tissue. Achieving directed differentiation to specific muscle or adipose tissue and simulating natural tissues is also the direction of our future efforts. The above results were added to the revised manuscript (as shown in **Extended Data Fig. 6f, g, 9e, f** and **Lines 197, 211**).

Response Figure 2: Investigation of cellular spatial organization origin via fluorescent protein-labeled reporter cell lines

a, Establishment of pgEpiSCs-mCherry fluorescent reporter cell lines and characterization of the adipogenic differentiation process. BODIPY represents the formation of LDs. Scale bar, 50 μm . **b**, Establishment of pgEpiSCs-NLS-GFP fluorescent reporter cell lines and characterization of the myogenic differentiation process. Myosin is marker for muscle fibers. Scale bar, 50 μm . **c**, Confocal fluorescence images of two-cell-type spheroids derived from pgMPCs-GFP and pgAPCs at day 15 and day 30. Scale bar, 100 μm . **d**, Confocal fluorescence images of three-cell-type spheroids derived from pgMPCs-GFP, pgAPCs-mCherry and pgVPCs at day 15 and day 30. Scale bar, 100 μm . **e**, Characterization of muscle/adipose spheroids at day 30. Green represents pgMPCs, Nile Red (red) represents pgADs. Scale bar, 100 μm (left) and 50 μm (right). **f**, Characterization of muscle/adipose/vascular spheroids at

day 30. Green represents pgMCs, red represents pgADs, yellow (CD31) represents ECs. Scale bar, 100 μm (left) and 50 μm (right). Similar results were obtained in three independent experiments.

Additionally, they must address whether the final tissue composition (e.g., muscle-to-fat ratio) is controllable through seeding ratios, and if so, provide supporting quantitative data.

Response: Thank you for your comment. Based on your suggestions, we provided additional experiments and results to demonstrated that the final tissue composition of the spheroids can be regulated by the cell seeding ratio, as illustrated by the following quantitative data:

For spheroids composed of two cell types, pgMPCs and pgAPCs were co-seeded at different ratios (pgMPCs: pgAPCs = 9:1, 8:2, 7:3, 5:5, 3:7, 2:8, 1:9). After 25 days of culture, bright-field imaging revealed their external morphology. Interestingly, we found that there were differences in the size of the spheroids inoculated at different ratios. The spheroids formed at ratios of 7:3, 5:5, and 3:7 were larger, while as the proportion of pgAPCs increased, the spheroids became smaller (2:8,1:9) (**Response Figure 3a**). Subsequently, samples were collected for sectioning and staining. After staining, the sections were imaged at the same fluorescence intensity, and the area fraction of fluorescence was quantified (**Response Figure 3b**). The results suggested that the proportion of different tissue components was basically positively correlated with the initial cell seeding rate (**Response Figure 3c**).

Response Figure 3: Characterization of spheroids formed at different seeding ratios of pgMPCs and pgAPCs.

a, Light microscopy images of spheroids formed from at different seeding ratios of pgMPCs and pgAPCs on day 25. Scale bar, 100 μ m. **b**, Representative Immunofluorescence images of the cross-sections from spheroids formed from at different seeding ratios of pgMPCs and pgAPCs on day 25. Myosin and BODIPY are markers for muscle fibers and adipocytes, respectively. Scale bar, 50 μ m. **c**, Quantification of the fluorescence area fractions of Myosin and BODIPY in spheroids with different seeding ratios of pgMPCs and pgAPCs. Error bars indicate \pm SD, n

= 3. * $p < 0.05$, ** $p < 0.01$, *** $p < 0.001$, **** $p < 0.0001$, and ns indicates $p \geq 0.05$. Similar results were obtained in three independent experiments.

For spheroids composed of three cell types, we co-seeded pgMPCs, pgAPCs, and vascular progenitor cells (pgVPCs) at three distinct ratios: 1:1:1, 1:1:2, and 2:2:1. Bright-field imaging revealed that spheroids formed at the 1:1:1 ratio exhibited the largest diameter; furthermore, prominent cavities were observed in spheroids with the 1:1:1 and 1:1:2 ratios, whereas spheroids with the 2:2:1 ratio displayed a more homogeneous texture (**Response Figure 4a**). Following sectioning and staining, samples were imaged at a consistent fluorescence intensity (**Response Figure 4b**), and the area fraction of fluorescence was quantified (**Response Figure 4c**). The results showed that the proportion of each tissue component varied across spheroids with different seeding ratios (**Response Figure 4c**).

In summary, regardless of whether two or three cell types were used, we have validated that the final tissue composition could be controlled by adjusting the seeding ratio, which provides a methodological reference for the nutritional modulation of future CM products.

These results were added to the revised manuscript (as shown in **Extended Data Fig. 8**, **Extended Data Fig. 10a-c** and **Lines 204-206, 223-224**). Thank you again for your comments and suggestions.

Response Figure 4: Characterization of spheroids formed at different seeding ratios of pgMPCs, pgAPCs, and pgVPCs.

a, Light microscopy images of spheroids formed from at different seeding ratios of pgMPCs, pgAPCs and pgVPCs on day 25. Scale bar, 100 μm . **b**, Representative Immunofluorescence images of the cross-sections from three-cell-type spheroids formed from at different seeding ratios. Myosin, BODIPY and CD31 are markers for muscle fibers, adipocytes and vascular endothelium, respectively. Scale bar, 50 μm . **c**, Quantification of the fluorescence area fractions of Myosin BODIPY and CD31 in three-cell-type spheroids with different seeding ratios. Error bars indicate \pm SD, n = 3. * p < 0.05, *** p < 0.001, ns indicates p \geq 0.05. Similar results were obtained in three independent experiments.

6. Evaluation of Nutritional Content:

The comparison of amino acid and fatty acid profiles between cultivated meat (CM) and traditional pork lacks sufficient normalization. The manuscript should provide absolute total amino acid and fatty acid contents per unit mass. Current figures (e.g., 5i, 5j) only show relative proportions within each group, which does not fully support the claim of nutritional similarity.

Response: We sincerely appreciate your insightful suggestion, which is critical for strengthening the nutritional equivalence argument of our CM product. According to your suggestion, we have supplemented the study with absolute quantification of total amino acids and fatty acids per gram of wet weight (ww) of CM, undifferentiated control cell mass pgEpiSCs and commercial pork (longissimus dorsi muscle). The result showed that the relative proportion of amino acid composition in pgCM was comparable to that of pork, albeit with a lower absolute content than pork. Regarding fatty acid composition, pgCM exhibited a higher proportion of polyunsaturated fatty acids (PUFAs) and lower monounsaturated fatty acids (MUFAs) than pork; additionally, its total fatty acids (TFAs), saturated fatty acids (SFAs), and PUFAs absolute

content exceeded those of pork and pgEpiSCs. These above results indicated that discrepancies remained of cell-cultured meat and conventional pork in their nutritional profiles.

This observation may be attributed to the seeding ratio of distinct cell components within the spheroids. Literature reports demonstrate that the intramuscular fat (IMF) content in the muscle of commercial pigs ranges from 1.5% to 3%^{1,2}, whereas that of local breeds or high-quality selectively bred varieties can reach up to 10%^{3,4}. The initial seeding ratio of the cell components in the spheroids was 1:1:1—this discrepancy in initial composition could be one of the contributing factors to the observed differences, which in turn influenced the composition and proportion of the final nutritional components. Nevertheless, as demonstrated in **Point 5**, the composition of the final tissue composition could be controllable through seeding ratios. This finding implies that differences could be addressed in subsequent studies by optimizing the culture system and modulating the initial seeding ratio of cells. Furthermore, we could regulate and further optimize the nutritional composition of the end product to cater to the demands of diverse consumer groups in the future. These results were added to the revised manuscript (as shown in **Extended Data Fig. 10d-g**, and **Lines 225-229**). At the same time, we have revised the claim of nutritional similarity (as shown in **Lines 29 and 63 and 243-244**).

Response Figure 5: Amino acid and fatty acid composition of pork, pgEpiSCs and pgCM. Error bars indicate \pm SD, $n = 3$. **** $p < 0.0001$, ns indicates $p \geq 0.05$. Similar results were obtained in three independent experiments.

References:

1. Font, I. F. M., Brun, A. & Gispert, M. Intramuscular fat content in different muscles, locations, weights and genotype-sexes and its prediction in live pigs with computed tomography. *Animal* **13**, 666-674 (2019).
2. Font, I. F. M., Tous, N., Esteve-Garcia, E. & Gispert, M. Do all the consumers accept marbling in the same way? The relationship between eating and visual acceptability of pork with different intramuscular fat content. *Meat Sci* **91**, 448-453 (2012).
3. Huang, Y. *et al.* A large-scale comparison of meat quality and intramuscular fatty acid composition among three Chinese indigenous pig breeds. *Meat Sci* **168**, 108182 (2020).
4. Serrano, M. P., Valencia, D. G., Nieto, M., Lázaro, R. & Mateos, G. G. Influence of sex and terminal sire line on performance and carcass and meat quality of Iberian pigs reared under intensive production systems. *Meat Sci* **78**, 420-428 (2008).

7. Texture and Flavor Similarity Claims:

The similarity in mechanical texture and flavor profiles between CM and pork requires more substantial explanation. What are the proportions of muscle, fat, and ECM in both CM and pork?

Response: Thanks for the comment and suggestion, as the texture and flavor similarity to native pork is a core criterion for the practical application of CM. According to your suggestion, we performed experiments to see the proportions of muscle, fat, and ECM in both CM and pork.

To enable a quantitative comparison of the proportions of these three tissue types in CM and native pork, we performed fluorescence staining and quantitative statistics of muscle (Myosin), fat (BODIPY), endothelium (CD31) and ECM (Collagen IV) of 1: 1: 1 spheroids and pig muscle tissue. These markers clearly indicate the composition of the tissue, although there is difference in morphology between the multitissue spheroids and native muscle tissue. Statistical analyses revealed that, in comparison with pig muscle tissue, 1: 1: 1 spheroid exhibit lower muscle content, higher fat and endothelial content, and slightly higher ECM content (**Response Figure 6**).

Response Figure 6: Characterization of 1: 1: 1 spheroids and pig muscle tissue.

a, Representative Immunofluorescence images of the cross-sections from muscle/adipose/vascular spheroids formed at 1: 1: 1 seeding ratios and pig muscle tissue. Myosin, BODIPY and CD31 are markers for muscle fibers, adipocytes and vesicular endothelium, respectively. Scale bar, 100 μ m (above) and 200 μ m (below). **b**, Fluorescence images of COIV from 1: 1: 1 spheroids and pig muscle tissue. Scale bar, 100 μ m (above) and 200 μ m (below). **c**, Quantification of the fluorescence area fractions of Myosin BODIPY, CD31, and COIV in 1: 1: 1 spheroids and pig muscle tissue. Error bars indicate \pm SD, n = 3. *** p < 0.001, **** p < 0.0001, ns indicates p \geq 0.05. Similar results were obtained in three independent experiments.

If they are significantly different, how is the texture similarity achieved? The current explanation that collagen and ECM contribute to texture must be supported with quantitative data.

Response: Thanks for the comment. Regarding the correlation between collagen, ECM and meat texture, we present the following elucidation, which incorporates a literature review and quantitative data support.

To further elucidate the origin of the textural properties of pgCM, sausages fabricated from undifferentiated pgEpiSCs cell mass were included as a negative control group. Relative to the negative control, pgCM sausages exhibited texture properties (hardness, cohesiveness,

gumminess, chewiness, and resilience) comparable to those of conventional pork sausages, and outperformed pgEpiSCs sausages across all these parameters. Notably, the springiness of pgCM sausages was better than that of pork sausages, which might be influenced by inherent cellular characteristics and the production method itself. Additionally, modifications were made to the sausage formulation and TPA-related parameters; detailed information is described in the methods of the revised manuscript (as shown in **Fig. 5m, n**, and **Lines 233-238, 587-598**).

The expression levels of ECM-related genes in multi-tissue spheroids derived from the three cell sources were significantly higher than those in the control pgEpiSCs, as revealed by qPCR (**Extended Data Fig. 9j**) and transcriptome analysis (**Fig. 5h, Extended Data Fig. 9k-n**). Notably, the expression levels of many marker genes related to ECM production (*COL1A1*, *COL3A1*, *COL4A1*, *FN1*, *LAMA4*) in multi-tissue spheroids were greater than those in spheroids derived from single cell type. Gene Ontology (GO) enrichment analysis of genes upregulated in the T3_3D group relative to other groups revealed that T3-specific biological pathways are predominantly enriched in cell fusion and ECM production, such as extracellular matrix structural constituent, cell adhesion molecule binding, structural constituent of muscle, collagen, extracellular matrix, laminin and fibronectin binding.

The above quantitative results indicate that multicell coculture promotes the generation of various tissues, as well as the accumulation of ECM and collagen, which could conferred textural attributes to pgCM, rendering it more comparable to native pork.

Notes: Appearance and textural properties of sausages from pork, pgEpiSCs and multiple tissue spheroids.

In addition, existing literature has demonstrated that collagen and ECM contribute to texture. The integrity of skeletal muscle is maintained by the intramuscular connective tissues (IMCTs) that are composed of extracellular matrix (ECM) molecules such as collagens, proteoglycans, and glycoproteins¹. The majority of the muscle's load-bearing capacity stems from ECM rather than the muscle fibers, highlighting the requirement for a robust supportive structure to sustain mature muscle cells². With animal growth, the structural integrity of IMCT increases; collagen fibrils within the endomysium associate more closely with each other, and the collagen fibers in the perimysium become increasingly thick and their wavy pattern grows more regular. These changes increase the mechanical strength of IMCT, contributing to the toughening of meat¹. Accordingly, recapitulating the mechanical properties of the native ECM is imperative for CM to attain the textural attributes of conventional meat³. This can be achieved either by utilizing scaffold materials with matching mechanical properties or by inducing cells to endogenously secrete their own ECM components^{4, 5, 6}. The scaffold-free methods based on organoids or spheroids do not introduce any exogenous scaffold materials but rely on the ECM

secreted by the cells themselves to bond the cells together, which may contribute to better organoleptic and nutritional similarity to conventional meat³.

In conclusion, the above results demonstrate that the proportion of ECM in CM was comparable to that in native meat, with similar textural properties observed between the two. And it also explained the contribution of collagen and ECM on the texture of meat.

Thank you again for your comments and questions. Hope our reply could address your concern.

References:

1. Gillies, A. R. & Lieber, R. L. Structure and function of the skeletal muscle extracellular matrix. *Muscle Nerve* **44**, 318-331 (2011).
2. Nishimura, T. Role of extracellular matrix in development of skeletal muscle and postmortem aging of meat. *Meat Sci* **109**, 48-55 (2015).
3. Bomkamp, C. *et al.* Scaffolding Biomaterials for 3D Cultivated Meat: Prospects and Challenges. *Adv Sci (Weinh)* **9**, e2102908 (2022).
4. Bodiou, V., Moutsatsou, P. & Post, M. J. Microcarriers for Upscaling Cultured Meat Production. *Front Nutr* **7**, 10 (2020).
5. Furuhashi, M. *et al.* Formation of contractile 3D bovine muscle tissue for construction of millimetre-thick cultured steak. *NPJ Sci Food* **5**, 6 (2021).
6. Ben-Arye, T. *et al.* Textured soy protein scaffolds enable the generation of three-dimensional bovine skeletal muscle tissue for cell-based meat. *Nature Food* **1** (2020).

8. PUFA/SFA Ratio Difference:

The PUFA/SFA ratio of CM is reported to be higher than that of traditional pork. The manuscript should explain this discrepancy, including whether it results from cell type composition, media composition, or other culture conditions.

Response: Thank you for the comment and suggestion. We have found that the Σ PUFA/ Σ SFA ratio in CM is approximately twice as high as that in conventional pork. To better explain this discrepancy, we analyzed the differences in fatty acid metabolism-related pathways between CM derived from multi tissue spheroids and commercial pork. The Gene Set Enrichment Analysis (GSEA) method is described in detail below:

Gene Set Enrichment Analysis (GSEA)

FastQC was used to conduct quality control (QC) analysis on the raw sequencing data obtained from pig muscle tissue and multitissue spheroids. Subsequently, Trimmomatic was employed for data filtering and adapter trimming to generate high-quality clean data. A pig genome index

was then constructed using STAR, followed by alignment of the filtered clean data to the pig reference genome. After alignment, featureCounts was utilized for gene quantification to generate a sample count matrix, which was further converted into a transcripts per million (TPM) expression matrix via a custom R script to support subsequent GSEA. Differential expression analysis was carried out using the DESeq2 package in R; additionally, GSEA was performed on the ranked list of genes using the fgsea package (R) to identify signaling pathways that were significantly activated in cell samples relative to native meat samples. Finally, visualization of all above-mentioned analysis results was implemented using the ggplot2 and pheatmap packages in R.

The results of GSEA showed that the signaling pathway related to unsaturated fatty acid (UFA) biosynthesis was significantly activated in multitissue spheroids compared with pig *longissimus dorsi* muscle. In the upper panel of GSEA Enrichment Plot (**Response Figure 7a**), the enrichment ranking metric curve of the GSEA enrichment analysis was entirely above the 0 scale, and the genes in this pathway were concentrated in the red region. With an FDR value of 0.004 (< 0.05) and a p -value < 0.05 , the result was statistically significant. Meanwhile, the NES value reached 6.875, and $|NES| > 1.5$, which indicated that the enrichment had biological significance and demonstrated that this signaling pathway was significantly activated. In the lower panel, the core genes with high contribution all fell in the region with large absolute values of \log_2FC , suggesting that these genes were the key driver genes for pathway enrichment.

An enrichment analysis was performed on these genes (**Response Figure 7b**), which are implicated in the biosynthesis of long-chain fatty acids and the regulation of lipid metabolism. Especially, *ELOVL2*, *ELOVL5*^{1, 2}, *FADS1*, *FADS2*^{3, 4}, *PECR*⁵ participate in the biosynthetic processes of PUFAs such as docosahexaenoic acid (DHA), eicosatetraenoic acid (EPA), and very long-chain fatty acids (VLCFAs), and are also associated with the maintenance of lipid homeostasis.

It is known that the fatty acid composition in cells *in vitro* is dependent on the culture medium and consequently differs from cells *in vivo*^{6, 7}. Therefore, differences between spheroids and native meat tissue were to be expected. Notably, antioxidants are commonly incorporated into *in vitro* cell culture systems to mitigate oxidative stress. In our experimental setup, the culture medium was supplemented with B27, ascorbic acid (VC), and GlutaMAX™

Supplement—all of which exhibit antioxidant properties and may potentially modulate the production of PUFA.

These above results and detailed information of the methods were added in the revised manuscript (as shown in **Fig. 5k, I** and **Lines 229-233, 521-531**).

Response Figure 7: Gene Set Enrichment Analysis (GSEA) of multi tissue spheroids and pig *longissimus dorsi* muscle.

a, GSEA Enrichment Plot of unsaturated fatty acid (UFA) biosynthesis. FDR, false discovery rate, ES, enrichment score, and NES, normalized enrichment score. **b**, Heatmap illustrating the expression of genes associated with UFA biosynthesis of T3_3D and pig muscle tissue. T3_3D, spheroids derived from three different cell type sources.

References:

1. Gonzalez-Soto, M. *et al.* Comparing Soy and Milk Protein Regulation of Hepatic Omega-3 Fatty Acid Biosynthesis. *Faseb j* **39**, e70805 (2025).
2. Blomquist, S. A. *et al.* The influence of FADS genetic variation and omega-3 fatty acid deficiency on cardiometabolic disease risk in a Mexican American population. *Front Nutr* **12**, 1538505 (2025).
3. Li, M. *et al.* Effects of Dietary n-6: n-3 PUFA Ratios on Lipid Levels and Fatty Acid Profile of Cherry Valley Ducks at 15-42 Days of Age. *J Agric Food Chem* **65**, 9995-10002 (2017).
4. White, P. J. *et al.* Transgenic ω-3 PUFA enrichment alters morphology and gene expression profile in adipose tissue of obese mice: Potential role for protectins. *Metabolism* **64**, 666-676 (2015).
5. Hua, T. *et al.* Studies of human 2,4-dienoyl CoA reductase shed new light on peroxisomal β-oxidation of unsaturated fatty acids. *J Biol Chem* **287**, 28956-28965 (2012).
6. Klatt, A. *et al.* Dynamically cultured, differentiated bovine adipose-derived stem cell spheroids as building blocks for biofabricating cultured fat. *Nature Communications* **15** (2024).
7. Else, P. L. The highly unnatural fatty acid profile of cells in culture. *Prog Lipid Res* **77**, 101017 (2020).

9. Scalability and Industrial Relevance:

The authors need to strengthen the discussion regarding how their strategy improves scalability and reproducibility of cultured meat. What are the implications for industrial applications, especially under suspension culture conditions?

Response: Thank you for the comment and suggestion. We further elaborated on the scalability and reproducibility of our strategy and summarized its advantages in the industrial production of cultivated meat as follows:

First of all, pgEpiSCs exhibit unlimited self-renewal capacity, enabling large-scale expansion in a serum-free culture system while fully maintaining their multi-lineage differentiation potential—thereby addressing the central bottleneck of "limited seed cell sources" in traditional cultured meat production^{1, 2}. Furthermore, the serum-free directed differentiation protocols for muscle, adipose, and endothelial cells established in this study are standardized through defined induction factors and precise temporal windows, effectively eliminating batch-to-batch variability associated with undefined components. When integrated with 3D suspension-based self-assembly platforms—such as stirred-tank bioreactors—this system enables volume-scalable cultivation, seamlessly transitioning from laboratory-scale flask to industrial-scale bioreactors. The above content demonstrate that our strategy proposed in this study exhibits favorable scalability.

What's more, the pgEpiSC line exhibits a stable genetic background and consistent differentiation potential³, while the directed differentiation system utilizes well-defined signaling pathway regulators without reliance on feeder cells or undefined serum components, thereby ensuring reproducible differentiation efficiency toward target cell populations—including muscle, adipose, and endothelial cells—across multiple independent experiments. Additionally, 3D self-assembly is mediated by intrinsic cell-cell recognition and adhesion mechanisms, eliminating the need for exogenous scaffolds or complex external forces⁴, which reduces artificial operational errors and ensures reproducible formation of multi-tissue spheroids with uniform structure and composition.

Suspension culture represents the cornerstone of industrial CM production due to its high space utilization and compatibility with process automation, and our strategy is fully aligned with this framework⁵. First, suspension-based 3D self-assembly overcomes the limitations of

2D adherent culture—such as low cell density and poor scalability—as well as those of scaffold-based systems, including high costs and potential safety risks associated with residual scaffold materials, thereby reducing production expenses and enhancing product safety^{4,6}. Second, the resulting multi-tissue spheroids, which integrate muscle, adipose, and vascular-like components, better simulate the compositional complexity and textural properties of natural meat. Moreover, the standardized protocol from pgEpiSCs expansion to multi-tissue spheroids support process optimization and automated control, including real-time monitoring of cell density, differentiation status, and spheroids formation in bioreactors, thereby meeting industrial requirements for high efficiency, stability, and large-scale production.

In summary, pgEpiSCs-derived multi-tissue self-assembly strategy in suspension culture is anticipated to overcome the constraints of scalability and reproducibility, thereby offering a feasible and industrializable technical pathway for high-quality cultured meat production.

Thank you again for your suggestion. The summarized discussion has been described in the revised manuscript (as shown in **Lines 289-296**).

References:

1. Jara, T. C. *et al.* Stem cell-based strategies and challenges for production of cultivated meat. *Nat Food* **4**, 841-853 (2023).
2. Martins, B. *et al.* Advances and Challenges in Cell Biology for Cultured Meat. *Annu Rev Anim Biosci* **12**, 345-368 (2024).
3. Zhi, M. *et al.* Generation and characterization of stable pig pregastrulation epiblast stem cell lines. *Cell Res* **32**, 383-400 (2022).
4. Bomkamp, C. *et al.* Scaffolding Biomaterials for 3D Cultivated Meat: Prospects and Challenges. *Adv Sci (Weinh)* **9**, e2102908 (2022).
5. Gu, H. *et al.* Scaling Cultured Meat: Challenges and Solutions for Affordable Mass Production. *Compr Rev Food Sci Food Saf* **24**, e70221 (2025).
6. Pasitka, L. *et al.* Empirical economic analysis shows cost-effective continuous manufacturing of cultivated chicken using animal-free medium. *Nature Food* **5** (2024).

10. Literature Context and Claims:

The authors claim that most previous research on cultured meat focused on muscle tissue, but this is not entirely accurate. Numerous studies have explored fat and other supporting tissues. The discussion section should acknowledge and properly cite relevant literature, such as:

- **Communications Biology 5, 927 (2022)**
- **Materials Today Bio 21, 100720 (2023)**
- **Food Chemistry 460, 140696 (2024)**
- **Nature Food 4, 35–50 (2023)**
- **Nature Communications 12, 5059 (2021)**

Response: Thanks for your suggestion. In the background section, we mentioned that current research has replicated the structure of meat by assembling various types of cells and cited relevant literature (as shown **Lines 54, 55**). We admit that we had an inappropriate discussion and have revised this part and cited the literature (as shown **Lines 265-267**).

11. Typographical Corrections:

In line 207 of the main text, the citation “Fig. g” should be corrected to “Fig. 5g.”

Response: Thanks to the reviewer for pointing out the problem. We apologize for the mistake we made in processing the images of the manuscript. Furthermore, the citation was corrected to “**Fig. 5g.**” (as shown in **Line 214**).

Reviewer #2 (Remarks to the Author):

The paper describes the generation of multitissue cell-cultivated meat using stable porcine epiblast stem cells. Researchers achieved multidirectional differentiation of these stem cells to produce various tissue types relevant for cultivated meat production. The authors previously already showed the differentiation of muscle tissue from pig epiblast stem cells (Generation of three-dimensional meat-like tissue from stable pig epiblast stem cells | Nature Communications) and extend this line of research here with a multi-cell type approach.

The presented paper claims to have developed a multi-tissue cultured meat product using porcine epiblast stem cells and 3D suspension culture for the application in cultured meat.

The publication contains several interesting routes of investigation that are relevant for the field of cultured meat and offer a good starting point for further research. The authors claimed to have developed a serum- and animal component free differentiation system that enables direct induction of muscle, adipose, and vascular endothelial cells.

Response: Thank you for the positive comments.

The main body of evidence the authors present is done in 2D, using also animal-derived components, and only limited evidence is presented for the 3D culture system (see the detailed feedback under “Major points”).

Response: Thanks for your comment and suggestion. We explained and discussed the animal-derived components, further provided additional evidence of 3D culture system, and refined the experimental protocol, hoping to address your concerns.

Major points**Claim 1: Serum- and animal component free differentiation system**

Several initial steps in the development to derive muscle and adipogenic progenitors used animal derived components (like feeder cells for expanding piEpiSCs and gelatin for pre-adipocyte differentiation).

Response: Thanks for your comment and question. In the experimental methods outlined in the manuscript, we referenced the utilization of two animal derived components: feeder cells and

gelatin. We agree that the development of a fully animal-free system is crucial for the commercialization of cultured meat.

In the entire differentiation system of pgEpiSCs, it is completely serum-free and free animal-components free. During the differentiation process, we compared gelatin-supplemented and gelatin-free systems and observed no significant differences between the two. We showed the morphological changes and proliferation capacity of differentiated cells at 24 h, 48 h, and 72 h post-passaging in gelatin and gelatin-free conditions, and indicated that gelatin is not an essential component for pgEpiSCs differentiation. We apologize for the mistake we made in the method section of the manuscript, and we have revised it accordingly. We modified this part in the method of the revised manuscript (as shown in **Lines 345 and 376**).

However, for the isolation and maintenance of pgEpiSCs, feeder cells and gelatin are necessary, as they play an important role in the self-renewal and quality maintenance of stem cells. While these components are indeed derived from animals, they represent a widely accepted "gold standard" in early-stage stem cell research due to their ability to provide essential paracrine signals and physical support. Notably, the stem cell culture system does not contain serum components. Although the acquisition and long-term cultivation of embryo-derived pluripotent stem cells under feeder-free conditions have not yet been fully achieved, these limitations could be overcome through progressive cell adaptation and optimization of the culture system. We have added a discussion on this limitation of the revised manuscript. (as shown in **Lines 262-264**).

Response Figure 1: The routine passaging of pgAPCs in the absence of gelatin

After 24 h, 48 h and 60 h of passaging pgAPCs under both gelatin-coated and gelatin-free conditions, cellular morphology and proliferation capacity were assessed. Scale bar, 200 μm .

While showing a completely animal-free process may be out of scope of this paper, those are relevant steps in the development of cultured meat and this limitation in the presented protocol has to be discussed and an outlook has to be given.

Response: Thanks for your suggestion. As you emphasized, moving towards a completely animal-free process is an important next step. Our study has demonstrated that the differentiation process of pgEpiSCs is serum- and animal-component-free. For the culture of pgEpiSCs, we are actively optimizing the medium and substrates to replace animal-derived factors. We have added a dedicated section in the "Discussion" to elaborate on these future directions (as shown in **Lines 261-264**).

Claim 2: Direct induction to muscle, adipose and vascular endothelial cells

In 2D, the authors show indeed directed differentiation from EpiSCs towards the 3 different tissues. There are, at least for adipogenic cells, similarities to in vitro derived adipocytes (from differentiated FAPs), but the EpiSC-derived adipocytes show lower maturity though in regards to the gene expression.

More importantly, little is shown about the size and maturity of the fat droplets in those cells. The fluorescent images in Fig. 1 are relatively small and it is hard to see if unilocular droplets are present. Here, better pictures including a FAP-derived control are advised to enable the reader to judge the differentiation efficiency better.

Response: Thank you for your positive comments and question. According to your suggestion, we have provided more magnified pictures (**Response Figure 2**). We observed that pgADs exhibit smaller lipid droplets (LD) compared to those derived from FAPs; however, both cell types contain multilocular lipid droplets. These results and were added in the revised manuscript (as shown in **Fig. 1g, h** and **Extended Data Fig. 2b, c**).

Regarding the maturity of adipocytes, study¹ reported that porcine preadipocytes cultured in vitro exhibit lipid accumulation. When compared to native adipose from pig, lipids within cultured porcine adipocytes were multilocular, unlike native adipose. A multilocular phenotype can signify adipocyte browning, indicating the presence of beige or brown fat^{2,3}. However, due

to the lack of a functional UCP1 gene, pigs reportedly have no brown adipose tissue, this multilocular lipid likely represents an immature white adipocyte phenotype. Furthermore, porcine FAPs isolated and cultured *in vitro* exhibit substantial disparities in adipogenic potential and lipid droplet morphology compared to mature intramuscular adipose tissue *in vivo*⁴. In white adipocytes differentiated from mouse⁵ and human⁶ pluripotent stem cells, the formation of such multilocular lipid droplets has also been observed, which may reflect the differences of adipogenesis between *in vivo* and *in vitro*.

In conclusion, the adipocytes generated via our differentiation protocol—whether derived from pgEpiSCs or pFAPs—exhibit substantial lipid accumulation; however, they may both represent incompletely mature adipocytes, with differences from native adipose tissue *in vivo*.

Response Figure 2: Characteristics identification of adipocytes derived from pgEpiSCs and pFAPs

a, Cell morphology and Oil Red O staining before and after the adipogenic differentiation of pgAPCs. Scale bar, 50 μ m. **b**, Fluorescently stained image of FABP4 and BODIPY in pgADs. Scale bar, 20 μ m. **c**, Morphology and Oil Red O Staining of proliferation and differentiation of

pFAPs and for different passages. Scale bar, 50 μm . **d**, BODIPY staining of LDs and immunostaining of FABP4 of adipocyte differentiated from P3 and P13. Green, BODIPY; red, FABP4; blue, DAPI. Scale bar, 20 μm .

Notes: The figure was cited from references¹.

Notes: The figure was cited from references⁴.

Notes: The figure was cited from references⁵.

Notes: The figure was cited from references⁶.

References:

- 1 Yuen, J. S. K., Jr. *et al.* Aggregating in vitro-grown adipocytes to produce macroscale cell-cultured fat tissue with tunable lipid compositions for food applications. *Elife* **12** (2023).
2. Zheng, Q. *et al.* Reconstitution of UCP1 using CRISPR/Cas9 in the white adipose tissue of pigs decreases fat deposition and improves thermogenic capacity. *Proc Natl Acad Sci U S A* **114**, E9474-e9482 (2017).
3. Berg, F., Gustafson, U. & Andersson, L. The uncoupling protein 1 gene (UCP1) is disrupted in the pig lineage: a genetic explanation for poor thermoregulation in piglets. *PLoS Genet* **2**, e129 (2006).
4. Wang, G. Q. *et al.* Mulberry 1-Deoxynojirimycin Inhibits Adipogenesis by Repression of the ERK/PPAR γ Signaling Pathway in Porcine Intramuscular Adipocytes. *J Agric Food Chem* **63**, 6212-6220 (2015).
5. Hudak, C. S. *et al.* Pref-1 Marks Very Early Mesenchymal Precursors Required for Adipose Tissue Development and Expansion. *Cell Reports* **8**, 678-687 (2014).
6. Karam, M., Younis, I., Elareer, N. R., Nasser, S. & Abdelalim, E. M. Scalable Generation of Mesenchymal Stem Cells and Adipocytes from Human Pluripotent Stem Cells. *Cells* **9** (2020).

Claim 3: Formation of tissue mimetic spheroids

While some data is shown on culturing adipogenic and myogenic progenitors in 3D (both alone and in co-culture), the data is missing more proof to be seen as a convincing model for a scalable culture:

Authors showed 30 days of culture with a final spheroid size of > 1mm and it is not believable that there was no cell death, e.g. in the form of a dead core, present. Just showing pictures from the surface of the structures is in this case not enough, cryosections would be needed to judge the inner core of structures for viability.

Response: Thanks for your comments and questions. We comprehensively analyzed and visualized the viability of the entire spheroids via live and dead cell staining, confocal imaging and flow cytometry analysis.

For spheroids formed by both alone and in co-culture progenitors, we conducted viability assays after culturing them for 30 days. After staining, confocal imaging was performed to further evaluate the cell viability throughout the spheroids: Calcein AM was used to label viable cells, while the Propidium Iodide (PI) stained the nuclei of dead cells. Meanwhile, flow cytometry analysis showed that the live cell rates were all above 90%. Collectively, the above results indicate that our 3D suspension culture system could serve as a scalable model for cells. These results were added to the revised manuscript (as shown in **Extended Data Fig. 5e, f, 6d, e, 9b, c, and Lines 197, 209**).

Response Figure 3: The identification of the activity of spheroids

Confocal imaging and flow cytometry analysis following Calcein AM/PI staining of muscle spheroid (a, b), fat spheroid (c, d), muscle-adipocyte spheroid (e, f) and muscle-adipose-vascular spheroid (g, h). Scale bar, 100 μ m.

While accumulation of fat droplets in the structures indicates successful adipogenic differentiation, the staining for myogenic differentiation is not convincing: maturing

skeletal differentiation leads to more alignment and cellular fusion, neither is shown in the presented data.

The co-culture (Figure 5) shows very limited differentiation, especially towards muscle. It doesn't seem convincing that all initially included cell types actually survived in the 3D structure or if one cell type may have overgrown the others. Again, while some marker expression is shown, no mature muscle (with alignment and fusion) is shown in those structures. It seems unlikely that those structures have a very different protein composition to undifferentiated material.

Response: Thanks for your comments and questions. Based on your suggestion, we repeated the experiment and supplemented it with additional experimental data to validate the maturity and structural characteristics of the spheroids.

We have supplemented additional representative stained section images, which clearly demonstrate the heterogeneous tissue distribution across different regions (**Response Figure 4a**). Moreover, to characterize the tissue composition of intact spheroids, we performed whole-spheroid staining followed by confocal imaging. Confocal micrographs revealed the presence of muscle, adipose, and endothelial tissues within the spheroid (**Response Figure 4b**). Additionally, F-actin staining delineated the cytoskeletal architecture of the entire spheroid (**Response Figure 4c**). These results were added to the revised manuscript (as shown in **Fig. 5f and Extended Fig. 9e**).

To quantitative comparison of muscle, fat, and endothelial tissue proportions, we performed fluorescence staining and quantitative statistics of muscle (Myosin), fat (BODIPY), endothelium (CD31) of 1: 1: 1 spheroids. Statistical analysis showed that the contents of the three types of cells are different, especially the endothelial cells were less, which may be related to the difference in growth rate of the different types of cells in this system (**Response Figure 4d**).

While the hallmarks of muscle cells and lipid droplet accumulation were observed, the structural organization of these constructs remains distinct from that of mature muscle tissue and native meat. Achieving directed differentiation into specific muscle or fat lineages and

simulate the structure of native meat is the direction of our future efforts. We also discussed the limitations and future prospects in the revised manuscript (as shown in **Lines 279-284**).

Response Figure 4: Identification of the multitissue spheroids from three types of progenitor cells

a, Section staining of the day 25 multitissue spheroid from seeding ratio of 1: 1: 1; MF20, BODIPY, and CD31 are muscle fiber, adipocyte and EC markers, respectively. Scale bar, 100 μm (above) and 50 μm (blow). **b**, Confocal imaging of the staining of the day 25 spheroid; Myosin, BODIPY, and CD31 respectively represent muscle fibers, adipocytes and ECs. Scale bar, 100 μm (above) and 50 μm (blow). **c**, F-actin staining of the 1: 1: 1 spheroid, Scale bar, 100 μm ; **d**, Quantification of the fluorescence area fractions of Myosin, BODIPY and CD31 in 1: 1: 1 spheroid. Error bars indicate \pm SD, n = 3. ** p < 0.01, *** p < 0.001, **** p < 0.0001. Similar results were obtained in three independent experiments.

Abstract and introduction make it sound as if the entire process could be done in 3D and in co-culture, albeit Fig. 5 indicates that only the last maturation step is done in 3D co-culture

Response: Thanks for your comment and suggestion. For the formation of the aggregates, we first differentiated each of the three cell types to the progenitor stage, followed by subjecting these progenitors to 3D suspension self-aggregation culture to form spheroids that undergo subsequent development and maturation. According to your suggestion, we have carefully revised the relevant content in the **Abstract** and **Introduction** sections regarding this aspect (as shown in **Lines 26 and 61-62**).

Claim 4: Replicates textural and nutritional value of conventional produced pork

It is interesting and promising to see the similarities between conventional pork, but as detailed above the limited differentiation efficiency shown in the 3D structures makes it hard to believe that the impact of the differentiation process is relevant. Here, an undifferentiated control cell mass would be needed to prove that the presented differentiation process actually makes an impact. It is possible that the hardness and guminess are more influenced by the production method itself. Here some more detail could help as well.

Response: Thanks for your comment and question. According to your suggestion, we performed experiment to compare the textural and nutritional value from undifferentiated control pgEpiSCs cell mass, pork and pgEpiSCs derived cell cultured meat (pgCM).

We fabricated sausages using control pgEpiSCs cell mass, and conducted comparative analyses with pgCM and conventional pork sausages. Additionally, the sausage formulation and equipment parameters were optimized and adjusted to better accommodate sausage texture characterization, as follows:

Texture Profile Analysis

TPA was measured by a texture analyzer (TA.XTplus, Stable Micro Systems Ltd (TPA mode, P/75 probe)). The cylindrical sausages samples, with a diameter of 15 mm and a height of 20 mm, were positioned at the center of the test platform, followed by TPA analysis. The parameters were set as follows: the pre-test speed was 1 mm/s, the test speed was 1 mm/s, and the post-test speed was 1 mm/s; trigger force set to 0.005 kg, trigger mode automatic; the degree of pressing down is 50%, the interval time is 5 s, and the reciprocation is carried out 2 times.

The Production of cultured and farmed pork sausage

Processing steps: Mincing → Ingredient weighing → Chopping/mixing → Filling/casing → Cooking → Cooling

Cooking conditions: temperature at 82°C, 45 min, casing diameter 15 mm (using 15 mL centrifuge tubes as an alternative)

The recipe for sausages

Ingredients	Conventional meat (%)	pgEpiSCs meat (%)	Cultured meat (%)
Pork (fat : lean=2.5:7.5)	75.00	50.00	50.00
pgEpiSCs	-	25.00	-
Cultured meat	-	-	25.00
Edible salt	1.60	1.60	1.60
Sodium tripolyphosphate	0.35	0.35	0.35
Ice water	15.00	15.00	15.00
Collagen	2.00	2.00	2.00
White pepper powder	0.12	0.12	0.12
Five-spice powder	0.05	0.05	0.05
Corn starch	5.00	5.00	5.00
Monascus red	0.012	0.012	0.012
White sugar	1.20	1.20	1.20
Total	100.332	100.332	100.332

The results demonstrated that pgCM sausages were comparable in appearance to both conventional pork sausages and pgEpiSCs-derived sausages (as shown in **Fig. 5m**). Textural profile analysis (TPA) revealed that pgCM exhibited similar hardness, cohesiveness, gumminess, chewiness, and resilience to conventional pork sausages, with all these attributes outperforming those of pgEpiSCs-derived sausages (as shown in **Fig. 5n**). Notably, both pgCM sausages and pgEpiSCs-derived sausages showed better springiness, which may be attributed to the intrinsic cellular properties. These findings indicate that cellular differentiation process exerts a critical impact on the texture of pgCM—differentiated 3D spheroids display a texture more analogous to that of pork than undifferentiated cell mass.

Response Figure 5: Appearance and textural properties of sausages from pork, pgEpiSCs and multiple tissue spheroids.

Furthermore, we have supplemented the study with absolute quantification of total amino acids and fatty acids per gram of wet weight (ww) of commercial pork (longissimus dorsi muscle) pgEpiSCs and pgCM. The result showed that the relative proportion of amino acid composition in pgCM was comparable to that of pork, albeit with a lower absolute content than pork but higher than pgEpiSCs. Regarding fatty acid composition, pgCM exhibited a higher proportion of polyunsaturated fatty acids (PUFAs) and lower monounsaturated fatty acids (MUFAs) than pork; additionally, its total fatty acids (TFAs), saturated fatty acids (SFAs), and PUFAs absolute content exceeded those of pork and pgEpiSCs.

These above results indicated that the presented differentiation process actually makes an impact about textural and nutritional value. The texture of cell-cultured meat exhibited a notable similarity to that of conventional pork; however, discrepancies remain in their nutritional profiles. These differences could be addressed in subsequent studies by optimizing the culture system and modulating the initial seeding ratio of cells.

These results along with the specific methodological details were added to the revised manuscript (as shown in **Fig. 5m, n, Extended Data Fig.10d-g** and **Lines 225–238, 588-598**).

At the same time, we have revised the claim of nutritional similarity (as shown in **Lines 29, 63 and 243-244**).

Response Figure 6: Amino acid and fatty acid composition of pork, pgEpiSCs and pgCM. Error bars indicate \pm SD, $n = 3$. **** $p < 0.0001$, ns indicates $p \geq 0.05$. Similar results were obtained in three independent experiments.

Also: while it is great to see the analytical specifications of their cultured meat product, it does not seem that actual tastings have taken place. Claims in regards to the taste therefore should be taken carefully.

Response: Thanks for your comment and suggestion. Furthermore, in line with the proposed recommendations, we revised the claims in regards to the taste, restricting our analysis to an objective assessment of the flavor compound composition of pgCM and pork sausages. Compared to pork sausages, pgCM sausages show a higher relative abundance of aldehydes and alcohols, whereas the levels of other compound classes are lower. The most pronounced differences are found in aldehyde and hydrocarbon content, suggesting distinct volatile profiles that may influence sensory characteristics. We have added these results to the revised manuscript (as shown in **Lines 238-241, Extended Data Fig.10h**).

Minor points

Fig. 1a typos (SB421542 instead of SB431542) and Roaiglitazone instead of Rosiglitazone

Response: Thanks to the reviewer for pointing out the problem. We apologize for the mistake we made in processing the images of the manuscript. Furthermore, the typos were corrected in **Fig. 1a**.

Fig 3a: A more detailed description of the final protocol for the co-culture (including details on the used basal medium and potential reasoning for the used growth factors) would be helpful. Generally, the method description is very superficial and does not enable replication. Since this is the most important novelty in the present publication, more detail would be expected

Response: Thanks for your questions. For the description of the method in Fig. 3a, we have already mentioned it in the **Method** section. Subsequently, we would elaborate on the materials employed and the experimental design in detail, while rectifying the errors identified therein.

2D monolayer coculture of pgEpiSCs-derived myoblast and preadipocyte

For coculture of pgEpiSCs-myoblast and preadipocyte, a suspension of myoblast and preadipocyte (at a ratio of 0:10, 1:9, 3:7, 5:5, 7:3, 9:1, 10:0) was seeded in twelve-well plates containing proliferation medium (BM (DMEM/F12 supplemented with 1% PS, 1% NEAA, 0.1 mM β -mercaptoethanol, 15% KOSR, and 200 μ M ascorbic acid) with 10 ng/ml IGF-1, 10 ng/ml HGF, 10 ng/ml FGF2 and 10 ng/ml hEGF). While cell fusion reached 90%, differentiation was initiated by switching to differentiation medium. The design of the two cultivation programs is presented as follows, (I) The cells were incubated in differentiation medium (ADM IV: MDM V=1:1) for 6 days, and in maintenance medium (ADM V: N2 medium=1:1) for 4 days. (II) The cells were maintained in ADM IV for 6 days, and maintained in ADM V for 3 days. The cells were maintained in a 5% CO₂ humidified incubator at 37 °C, with medium exchanges every 2 days.

Because the time corresponding to each stage of myogenic and adipogenic differentiation is different, regarding the used growth factors, it mainly corresponds to stage IV- stage V from myogenic and stage III- stage V adipogenic differentiation of pgEpiSCs. We designed two experimental schemes above, and the quantitative data proved that scheme (I) was superior, which was adopted for subsequent experiments and aligned with Fig. 3a. In addition, we have made some revisions to Fig. 3a and the method (as shown in **Lines 429, 432-433**).

Extended Fig 4: What does the edible scaffold consist of?

Response: Thanks for your suggestion. We sincerely apologize for the oversight in describing this section. For the fabrication of the 3D scaffold, we employed the method detailed in our previously published work. The scaffold materials and design are described in detail below:

Fabrication of Animal-Component-Free 3D Edible Plant-Based Polysaccharide Scaffolds

The plant-based 3D edible scaffolds were fabricated via an ion-crosslinking solvent displacement method as previously described¹. In brief, a homogeneous mixture was prepared by combining 3% (w/v) konjac glucomannan (KGM), and 3% (w/v) sodium alginate (SA) solution at a ratio of 1: 1, followed by the addition of 2% (w/v) CaCl₂ (in 75% ethanol aqueous solution). Then, the mixture was transferred into a 24-well plate (0.5 g/well) and pre-frozen at -20 °C for 12 h. The scaffolds were subsequently obtained by introducing 2% (w/v) CaCl₂ aqueous ethanol solution into each well and allowing solvent displacement to proceed for 24 h.

These above have been thoroughly elaborated in the methods of the revised manuscript (as shown in Lines 435-441).

Also, not sure if the culture in the scaffold adds anything to the presented story, as authors mention themselves that 2D expansion and scaffolds are not a viable option for the future

Response: Thanks for your comment and question. Regarding the contribution of scaffold-based culture to the presented story, the following explanations is provided: while 2D culture effectively supports the directed differentiation of pgEpiSCs into muscle, adipose, and vascular endothelial lineages, 3D scaffolds provide a microenvironment that enables the spatial organization of these three cell types into tissue-like multicellular structures—a critical step unattainable with 2D monolayers. Our work on 3D scaffolds serves as a transitional approach between 2D monolayer and 3D suspension culture systems. The progression from 2D culture to 3D scaffold-based culture, and further to 3D suspension self-assembly culture, represents a process of gradual optimization. Concurrently, this progression validates the capacity of our differentiated cells to undergo co-culture across distinct environmental contexts.

Discussion: The discussion missed an evaluation of the current limitation of their approach. As said, this is a promising starting point for cultured meat research, but

honest discussion of limitations (like that initial steps are done in 2D and the use of very expensive and not food safe materials) should be discussed and an outlook should be given IF the authors have reason to believe that those limitations can be overcome.

Response: Thanks for your comment and suggestion. According to your suggestion, we have discussed the limitations of our study honestly and given an outlook.

Firstly, successfully directed the differentiation of a single pgEpiSCs line into skeletal muscle, adipose, and vascular endothelial cells with a serum- and animal component-free differentiation system. Although the acquisition and long-term cultivation of embryo-derived pluripotent stem cells under feeder-free conditions have not yet been fully achieved, these limitations could be overcome through progressive cell adaptation and optimization of the culture system.

In addition, we employed a simple yet efficient 3D suspension culture system for the cultivation of multiple cell types, which spontaneously self-assemble into spheroids by relying on their endogenous ECM, forming multitissue architectures that effectively enhanced cellular proliferation efficiency and tissue complexity. It also facilitates large-scale cell culture and expansion, and helps seamlessly adapt to bioreactor systems. However, the requirement for an initial 2D differentiation step from pgEpiSCs to progenitor cells prior to 3D coculture results in complex procedures, necessitating further simplification and optimization. Overall, it is expected to overcome the limitations of scalability and repeatability, thereby providing a feasible and industrializable CM production technology approach. We also discussed the limitations and future prospects in the revised manuscript (as shown in **Lines 261-264, 296-299**).

Reviewer #3 (Remarks to the Author):

Response: Thank you very much for co-reviewing this manuscript.

Other changes:

- Following the reviewers' suggestions, we updated the pertinent experimental results and revised the figures, as well as further refined the methods section based on the newly added experiments' results, which are highlighted in red in the revised manuscript.
- Following the reviewers' comments, we further revised the discussion section.
- As a result of responding reviewers' comments, the references have been rearranged with red highlights in the revised manuscript.

We would like to thank the reviewers once again for their time and effort in providing helpful suggestions on our manuscript.